

# Erosional response of granular material in landscape models

Riccardo Reitano[1], Claudio Faccenna[1,2], Francesca Funiciello[1], Fabio Corbi[1,3,4], Sean D. Willett[5]

[1]Dipartimento di Scienze, Laboratory of Experimental Tectonics, Università "Roma Tre", Rome, 00146, Italy
[2]Department of Geological Sciences, Jackson School of Geosciences, The University of Texas at Austin, Austin, TX, USA
[3]Freie Universitat Berlin, Berlin, Germany
[4]Helmholtz Centre Potsdam - GFZ German Research Centre for Geosciences, Potsdam, Germany
[5]Department of Earth Sciences, ETH-Zürich, Sonneggstrasse 5, 8092 Zürich, Switzerland

*Correspondence to*: Riccardo Reitano (r.reitano@gmail.com)

**Abstract.** Tectonics and erosion/sedimentation are the main processes responsible for shaping the earth surface. The link
between these processes has strong influence on the evolution of landscapes. One of the tools we have for investigating coupled
process models is analogue modeling. Here we contribute to the utility of this tool by presenting laboratory-scaled analogue
models of erosion. We explore the erosional response of different materials to imposed boundary conditions, trying to find the
composite material that best mimics the behaviour of the natural prototype. The models recreate conditions in which tectonic
uplift is no longer active, but there is an imposed fixed slope. On this slope the erosion is triggered by precipitation and gravity,
with the formation of channels in valleys and diffusion on hillslope that are function of the analogue material. Using Digital
Elevation Models (DEMs) and laser-scan correlation technique, we show model evolution and measure sediment discharge
rates. We propose three main components of our analogue material (silica powder, glass microbeads and PVC powder) and
we investigate how different proportions of these components affect the model evolution and the development of landscapes.
We find that silica powder is the main responsible for creating a realistic landscape in laboratory. Furthermore, we find that
varying the concentration of silica powder between 40 wt.% and 50 wt.% (with glass microbeads and PVC powder in the range
35-40 wt.% and 15-20 wt.%, respectively) results in metrics and morphologies that are comparable with those from natural
prototypes.

## 1 Introduction

Whenever tectonics creates topography, erosion and surface processes act in response to the imposed gradient, tending to wipe
it out. During the last decades a strong theoretical background has been built based on field- and analytical observations (e.g.
Howard, 1994; Kirby and Whipple, 2001, 2012; Tucker and Whipple, 2002; Whipple et al., 1999; Whipple and Tucker, 1999,
2002) but since natural observations provide only a snapshot of processes acting at different timescales (e.g. Castillo et al.,
2014; Cyr et al., 2014; Pederson and Tressler, 2012; Sembroni et al., 2016; Vanacker et al., 2015) a quantitative framing of
the existing feedbacks between surface processes and tectonics in modifying the topography remain a difficult task. Analytical,
numerical and analogue models are often used by tectonic geomorphologists to improve the understanding of the feedbacks
between tectonics and surface processes. Numerical models have the advantage of a straightforward quantitative and



parametric approach and the possibility to be conducted with precise boundary conditions. Previous numerical and analytical focused on: the mathematical implementation in solving the stream power law (Braun and Willett, 2013); the interaction between surface processes and the velocity discontinuities bounding a double-verging orogenic wedge (Braun and Yamato,

2010); the coupling between climate, erosion and tectonics (e.g. Batt and Braun, 1999; Beaumont et al., 1992, 2001, 2004; Jamieson et al., 2004; Ueda et al., 2015; Whipple and Meade, 2004); the interaction between surface processes and multilayer folding systems (Collignon et al., 2014);the role of orographically enhanced precipitation in a double-verging 2D model (Willett et al., 1993; Willett, 1999); the control exerted by tectonic strain (Castelltort et al., 2012; Duvall and Tucker, 2015; Goren et al., 2015) or interaction with tectonics in three dimensions (Ueda et al., 2015). Nevertheless, the computational

capacities necessary to realistically simulate geologic features coupled with erosion using complex rheologies and/or three-dimensional settings remain limited.

Analog models can in principle overcome these limitations, allowing also a useful direct control on the evolution of the studied physical process. The coupling between tectonics and surface processes has been investigated using sand-box like analogue apparati including removal of material by hand or with a vacuum cleaner (e.g. Hoth et al., 2006; Konstantinovskaya and

Malavieille, 2011; Malavieille et al., 1993; Mulugeta and Koyi, 1987) or the application of a defined precipitation rate for a spontaneously-developing landscape (e.g. Bonnet, 2009; Lague et al., 2003; Schumm and Parker, 1973; Tejedor et al., 2017). The former models can be considered as "dry" models, where no water is added to the system. In this case, brittle wedges are typically built, and after a certain amount of shortening the material is removed from the wedge and sifted in the lower basins. The latter models can be considered as "wet" because water is added to the system and is responsible for erosion, transport

and sedimentation. Wet models mainly focus on surface uplifting/lowering (e.g. Bonnet and Crave, 2003; Hasbargen and Paola, 2000; Schumm and Rea, 1995; Singh et al., 2015) or in creating topography by horizontal advection (e.g. Graveleau et al., 2015; Graveleau and Dominguez, 2008; Viaplana-Muzas et al., 2019). Different granular materials have been used, like dry quartz sand (e.g. Persson et al., 2004), silica powder (e.g. Bonnet, 2009), mica flakes (e.g. Storti et al., 2000), glass microbeads (e.g. Konstantinovskaya and Malavieille, 2011), natural loess (e.g. Lague et al., 2003), walnut shells (e.g. Cruz et

al., 2008) or *ad-hoc* composite materials (e.g. Graveleau et al., 2011, 2015). These materials show different behavior in response to the external forcing, and their characterization is a key ingredient for scaling analogue models. The link between the properties of the materials and their tuning on the morphological response is not well defined yet. Even if some recent efforts have been done on pure materials (Graveleau et al., 2011), an *excursus* on the role played by the concentration of every component in a composite material is still lacking.

Here we focus on this argument exploring how different concentrations of granular materials influence the erosional, physical and mechanical response of several composite materials with the overarching goal of finding an analogue material that best mimics the erosional behavior of the natural prototype. We also focus on mechanical properties of the same materials, which will be involved in future projects.



## 2 Experimental Approach

In this study we analyzed four different brittle granular materials to be used as rock analogues for the upper crust: Silica Powder (SP), Glass Microbeads (GM), Crushed Quartz (CQ), PolyVinyl Chloride powder (PVC). These pure materials are suited in five different mixes, in different proportions, while one single material (SP) is tested on its own (Tab. 1). The selection has fallen on granular materials because:

   a) They have the proper physical properties to simulate downscaled crustal rock behavior under laboratory conditions
in a natural gravity field (e.g. Davis et al., 1983; Lallemand et al., 1994; Mulugeta and Koyi, 1987; Schreurs et al., 2001, 2006, 2016; Storti et al., 2000; Storti and McClay, 1995). As a matter of fact, they obey to Mohr-Coulomb failure criterion, showing strain-hardening prior to failure at peak strength and strain-softening until a stable value is reached (stable friction) (Schreurs et al., 2006);

   b) they reproduce reasonable geomorphic features due to developing of erosion/sedimentation processes like incision,
mass wasting and diffusive erosion, transport and sedimentation, although important differences in behavior and characteristics have emerged (Graveleau et al., 2011, 2015; Graveleau and Dominguez, 2008; Viaplana-Muzas et al., 2019).

In the following, we describe a) scaling to the natural prototype; b) geotechnical characterization of the materials (including geometrical and physical/chemical properties, frictional properties and permeability); c) erosional characterization.

## 2.1 Mechanical properties

Here we describe the mechanical properties of four granular materials mixed in different proportions. The aim of this analysis is bivalent: to study how mechanical properties affect erosion style and to define properties that will be used in future works, where the materials will be involved in experiments where active tectonics is present. In five experiments the analogue material are mixtures of the previous materials at various percentage (CM1-5, Tab. 1). In Graveleau et al. (2011), the authors display
how these different pure materials (excepting CQ, which was not considered in their work) show advantages and disadvantages in responding to erosion and sedimentation, in terms of morphological features developed, and brittle behavior. For example, GM and PVC produce high scarps with almost no channelization when erosion is applied, but realistically reproduce the brittle rheology of the upper crust (Graveleau et al., 2011). SP morphologies scale well with natural landscapes, but the higher strength of the powder induces unreliable structures under deformation. To overcome the above limitations of materials used as single
"ingredient", the authors suggested that a mixture of these three granular materials can be the most appropriate choice. Following Graveleau et al. (2011), here we focus on these mixes rather than pure materials. The latter are still analyzed, highlighting their role in the mix.

We measured geometrical, physical and frictional properties like grain size and shape, density, porosity and permeability, internal friction angle and cohesion. We measured frictional properties of experimental granular material (internal friction
coefficient $\mu$ and cohesion $C$) with a Casagrande shear box. We performed tests for peak and stable friction at variable normal





stresses. Density has been measured with a helium pycnometer. The grain size has been estimated via a series of sieves of decreasing openings dimension (from 250 µm to 45 µm). The material passing the 45 µm sieve has been analyzed using sedimentation in a distilled-water tank, with a hydraulic pump for recirculation of water and a thermal control for estimation of water density. We also used a laser diffractometer for checking the reliability of the previous measurements. A qualitative

analysis has been carried out using a Scanning Electron Micrograph (SEM) for the shape of grains and composition.

### 2.1.1 Geometrical and physical/chemical properties

The materials physical properties like grain size, density, porosity and permeability are listed in Table 2. The SP is a very fine powder ($D_{50} = 20$ µm), with clasts of different shape and size (Fig. 1): the smallest ones, are elongated and may lay on bigger clast, with a very high roughness. These characteristics require a careful use of this powder, due to the danger for the respiratory

system. The density is the highest among the studied components ($2661 \pm 1$ kg m$^{-3}$). We obtain composition of ~95% $SiO_2$, ~3% $Al_2O_3$, ~1% $K_2O$ and < 1% of $Na_2O$, $MgO$ and $CaO$ (Fig. 2). The CQ has bigger dimensions with respect to SP ($D_{50} = 87$ µm), with medium sphericity of the clasts and high roughness. The composition is very similar to silica powder (with ~0.5% of $FeO$) and so is the density, slightly lower ($2588 \pm 1$ kg m$^{-3}$). GM ($D_{50} = 98$ µm) have a very high sphericity and a very low roughness, where density ($2452 \pm 1$ kg m$^{-3}$) is lower than the CQ. The GM qualitative composition is made of ~69% of $SiO_2$,

~15% of $Na_2O$, ~10% of $CaO$, ~4% of $MgO$, ~1% of $Al_2O_3$ and < 1% of $K_2O$ and fluorine. Finally, the PVC ($D_{50} = 181$ µm) has a similar shape respect to glass microbeads but less uniform. The density is the lowest among the components ($1402 \pm 1$ kg m$^{-3}$), and this has a strong effect on the erosive properties of the material, as we will show afterwards. We did not perform SEM measurements of the PVC, due to the complexity of its chemical composition (($C_2H_3Cl)_n$). The grain size, density and shape of grains in the mixes are function of the percentage of every single material that form it (Tab. 2).

### 2.1.2 Frictional properties

A good crustal analog material must fail following the Mohr-Coulomb failure criterion (e.g. Davis et al., 1983; Davy and Cobbold, 1991; Krantz, 1991):

$$\tau = \mu\sigma + C \tag{1}$$

where $\tau$ is the shear stress corresponding to the normal stress $\sigma$ on the failure plane, and $\mu$ is coefficient of internal friction

defined as $\mu = \tan(\varphi)$, with $\varphi$ angle of internal friction. For geomorphic experiments, where water is added to the system, parameters like $\mu$ and $C$ strongly control the evolution of the experiment, because they change with the amount of water. The brittle granular materials typically used in laboratory models have low elasticity and undergo plastic deformation when their yield strength is reached, sliding along discrete fault-analog planes (e.g. Panien et al., 2006; Ritter et al., 2018; Rossi and Storti, 2003; Schreurs et al., 2016). Deforming granular materials satisfy Mohr-Coulomb criterion (Eq. 7), that highlights the

relationship between shear stress $\tau$ and normal stress $\sigma$ on the failure plane. The criterion typically shows a linear trend for $\sigma$ in the order of kPa and MPa, but a convex-outward envelope for normal stresses in the order of hundreds of Pa or lower (Schellart, 2000). In this work the normal stress applied are in the range 25-200 kPa. For every test we defined peak friction



(the first peak in the shear curve (Fig. 3) reflecting hardening-weakening during strain localization and then fault initiation) and stable friction (the plateau after peak friction representing friction during sliding (e.g. Montanari et al., 2017). A shear box has been used. It consists of a steel box (Rossi and Storti, 2003) split across its middle in two small blocks with an area of 6×6 cm$^2$ each. The bottom block is fixed, while the top block moves at constant velocity of 0.165 mm min$^{-1}$. Two dynamometers record horizontal and vertical displacement. The tests were made in water saturated condition. From every measurement we defined the material internal friction coefficient ($\mu$, slope) and cohesion ($C$, intercept) for peak and stable friction. Results for every material and mix are listed in Table 2. Between pure materials SP shows the highest values for $\varphi$ and $C$ (33-40° and 0-8.5 kPa respectively). For GM internal friction angle is about 23-25° and the cohesion is close to 0 or negative, so is not considered in this analysis. PVC shows the same conditions for $C$, with an internal friction angle between 25° and 32°. The frictional properties for the CQ are similar to SP, with $\varphi$ between 30-33° and $C$ between 4.5-6.6 kPa. The mixes show a strong variability for $\varphi$ and $C$, with average values about 31° and 6 kPa, respectively. The values of $\varphi$ and $C$ do not highlight the increasing of SP concentration from CM2 to CM5, while the mechanical properties seem to show a common trend.

### 2.1.3 Porosity and permeability

Porosity is defined as the ratio between the volume of voids or pore spaces ($V_V$) and the total volume ($V_T$):

$$\gamma = \frac{V_V}{V_T} \tag{2}$$

The porosity has been computed measuring the volumetric change of a weighted amount of material respect to an ideal condition where no pores are in the samples. We used a vibrating plate looking for the best compaction of material and measured the variation in volume. We used this technique to draw close to the experimental conditions. This procedure has been repeated several times for each composite material. Unfortunately, porosity is dependent on the handling technique, it is thus impossible to precisely control the porosity of the materials during preparation. The values of porosity for single materials and mixes are shown in Table 2. GM shows the lowest porosity (0.26) between pure materials while SP and CQ the highest one (0.36-0.37, respectively). As far as the mixes is concerned, only CM1 shows values higher than 0.40, while porosity is around 0.30 from CM2 to CM5.

Permeability represents the ability of a material to transmit fluids. This property has been tested using an oedometer and measuring the velocity of water flowing through the sample. This parameter is essential in controlling the evolution of our models, as it will be explained later in the text. SP has the lowest permeability (3.56·10$^{-14}$ m$^2$) while GM has the highest permeability (2.87·10$^{-12}$ m$^2$). CM1 shows the highest permeability between the various mixes (7.42·10$^{-13}$ m$^2$). From CM2 to CM5 (from 40 and 70 wt.% of SP) the permeability slightly decreases. However, CM3 and CM5 do not strictly follow this trend: the former has the lowest permeability observed in mixes (9.25·10$^{-14}$ m$^2$), while the latter show a value comparable with CM2 and CM4 but slightly higher (4.06·10$^{-13}$ m$^2$). The permeability values for mixes are then in the order of 10$^{-13}$ m$^2$.





## 2.2 Erosional characterization

### 2.2.1 Erosion laws and erosive properties

In a stream channel, the relationship between channel slope $S$ and contributing area $A$ is often expressed through Flint's Law (Flint, 1974), and takes the form

$$S = k_s A^{-\theta} \qquad (3)$$

where $k_s$ and $\theta$ are the steepness and concavity index respectively. The most common erosion law, consistent with slope-area scaling, for channelized processes is a power law function of the contributing area $A$ and surface gradient $S$, defined as the

"stream power" law (e.g. Howard, 1994; Tucker and Whipple, 2002; Whipple and Tucker, 1999)

$$\frac{dz}{dt} = KA^m S^n \qquad (4)$$

where $z$ is the elevation of the stream channel (i.e. $dz/dt$ elevation trough time), $K$ is the erosional constant (bound up to the erosional efficiency) that contains information about lithology, climate and channel geometry (Howard et al., 1994; Whipple and Tucker, 1999), while $m$ and $n$ are two positive dimensionless exponents, with the ratio $m/n$ (i.e. the concavity index $\theta$) that

typically falls between 0.4 and 0.7 (Tucker and Whipple, 2002). This model is better known as a "detachment-limited" stream power model, because in tectonically active regions or in condition of steep topography, the channel erosive power is high and limited by its capacity to detach particles from the bedrock (Tucker and Whipple, 2002; Whipple and Tucker, 2002). It is possible to rewrite Eq. (10) in terms of distance $x$ along the stream using Hack's Law (Hack, 1957)

$$A = k_a x^H \qquad (5)$$

where $k_a$ is a scaling coefficient and $H$ is the reciprocal of the Hack's exponent. Combining Eq. (10) and Eq. (11) we obtain

$$\frac{dz}{dt} = \kappa x^{Hm} S^n \qquad (6)$$

where $\kappa = K k_a^m$. In our experiments, $K$, $\kappa$, $m$, $n$, $H$ should be constant (for the same experiment), due to the homogeneous lithology and constant precipitation rate.

The analogue material should erode via linear erosion (i.e., advection in valleys), diffusion (i.e., on hillslopes) and mass

wasting. This requires that precipitation collects in surface drainage networks, with branching channels in order to be consistent with Hack's Law and slope-area scaling. For the erosional behavior of the composite material, the ratio between precipitation rate and infiltration capacity appears to be the main factor controlling the geomorphological response. If the precipitation rate is higher than the infiltration capacity, the model can develop surface runoff. Otherwise, the water flows through the inner part of the model, inducing fast erosion through discrete and rapid events. Fine sand or powders have been typically used for

geomorphic experiments (e.g. Babault et al., 2005; Hasbargen and Paola, 2000), so that runoff could develop (i.e., low infiltration capacity due to the grain size). Nevertheless, different materials exhibit different emergent morphological characteristics when precipitation is imposed. Among the pure materials presented above, only SP (or a mix with SP) successfully reproduces linear incision (e.g. Bonnet and Crave, 2006; Graveleau et al., 2011; Schumm and Parker, 1973; Tejedor et al., 2017), while GM, PVC (Graveleau et al., 2011) and CQ erode mostly by diffusion or mass wasting.



## 2.3 Experimental Setup for erosional characterization

For studying the composite materials response to applied boundary conditions (precipitation rate and slope), we developed a new experimental setup depositing the material into a box on an inclined plane under rainfall (precipitation). Both the initial slope for the apparatus and the precipitation rate are kept constant. No kinematic conditions of sidewalls are applied: no active tectonic is thus reproduced in our model.

The experimental setup is made of three different devices (Fig. 4): the box, the rainfall- and the monitoring systems. The material box controls the imposed initial slope, while the rainfall system triggers surface erosion. The evolution of the model is recorded with digital images and a laser scanner. The only forcing applied to the models is due to the gravity acceleration, that allows for the erosion triggered by slope and rainfall.

### 2.3.1 Box

The box is a plexiglass tank $0.35\times0.3\times0.05$ m$^3$, filled with the experimental material, water saturated (about 25 wt.%). After pouring the material into the box and leveling, it is left flat at least 12 hours, to avoid prior deformations. The slope of the box is then fixed at $15°$, in analogy to what has been done in Graveleau et al. (2011).

### 2.3.2 Rainfall system

Three nozzles fixed to an aluminum frame produce a high-density fog in which the droplet size is small enough ($\leq 100$ µm) to avoid rainsplash erosion (e.g. Bonnet and Crave, 2006; Graveleau et al., 2012; Lague et al., 2003; Viaplana-Muzas et al., 2015). The precipitation rate is controlled by both water pressure and the number of sprinklers. In our models the precipitation rate is fixed to 25-30 mm h$^{-1}$. The configuration allows for a homogeneous droplets distribution with a spatial variation of about 20%. The precipitation rate induces surface runoff, channel incision and gravity-driven processes that are responsible for the erosion of the model.

### 2.3.3 Monitoring system

Each experiment is recorded using one camera and a laser scanner. The camera records the model evolution in oblique view. The laser horizontal and vertical resolution are 0.05 mm and 0.07 mm, respectively. The scans are converted in digital elevation models (DEMs) using MATLAB. DEMs are analyzed with TopoToolbox (Schwanghart and Scherler, 2014) for geomorphological quantifications. Erosion and sediment discharge are computed with *ad-hoc* MATLAB algorithms. Stopping the rainfall and letting the surface dry is required to avoid distortions during the laser scan. At the beginning of the experiment pictures and scans are taken every 15 min, then every 30 min and 60 min, depending on model evolution rate.





## 2.4 Scaling analysis

An analogue model should be scaled by its geometry, kinematic, dynamic and rheological properties (e.g. Hubbert, 1937; Ramberg, 1981). The length scaling factor ($h* = h_{model}/h_{nature}$) commonly used in sandbox experiments for deformation in the

upper crust with granular material ranges between $10^{-5}$ and $10^{-6}$ (e.g. Cruz et al., 2008; Davy and Cobbold, 1991; Konstantinovskaya and Malavieille, 2011; Persson et al., 2004; Storti et al., 2000). Hence, 1 cm in the model may be in the range of 1 – 10 km in nature. The gravitational acceleration model-to-nature ratio is set to 1 ($g* = g_{model}/g_{nature}$) working in the natural gravitational field. The density of the dry quartz sand or corundum sand employed in literature defines the model-to-nature ratio for density ($\rho* = \rho_{model}/\rho_{nature}$) to be around 0.5 – 0.6. Since the dimensionless coefficient of internal friction is

very similar between the analogue material and the natural crustal rocks, the cohesion/body forces scaling factor can be expressed as

$$\sigma^* = \rho^* g^* h^* \tag{7}$$

Typical values of this scaling factor are in the order of $10^{-6}$, so that 1 Pa in the model would correspond to about 1 MPa in nature (e.g. Buiter, 2012; Graveleau et al., 2012).

The classical approach of analog modelling of convergent wedges is time independent (i.e., the evolution of the model is independent from convergence rate). Here, together with compressional tectonics we also model erosion with the perspective of implementing the tested materials in analog models where tectonics and erosion are coupled. Therefore, following Willett (1999), we introduce a time scaling factor $t^*$ which is the ratio between the mass flux added to the system $F_{in}$ over the mass flux removed $F_{out}$. $F_{in}$ is defined as follow:

$$F_{in} = v_c h \tag{8}$$

where $v_c$ and $h$ are the convergence velocity and initial thickness of the layers considered, respectively. $F_{out}$, is defined as:

$$F_{out} = 4KL^2 \tag{9}$$

where $K$ is a constant proportional to bedrock incision efficiency and precipitation rate (Eq. 4 and 6) and $L$ is the wedge width. Assuming $m = 0.5$ and $n = 1$ in Eq. (4) (e.g. Whipple and Tucker, 1999), the $K$ parameter has the dimension of time$^{-1}$. Therefore,

the mass balance can be expressed as the ratio between these two fluxes:

$$M_b = \frac{4kL^2}{v_c h} \tag{10}$$

$M_b$ is a dimensionless number, so keeping it the same in the experiment as in nature (in the same gravity field) it is possible to derive the scaling factor for time rewriting Eq. (4) as

$$\frac{4k^* L^*}{v_c^*} = 1 \tag{11}$$

and considering that $k^*$ has the dimension $t^{-1}$, so that

$$t^* = \frac{4L^*}{v_c^*} \tag{12}$$

where * marks the model over nature dimensionless ration for every quantity, as defined before. With this scaling factor, considering $L^* = 10^{-5} – 10^{-6}$ and $v_c^* = 8 \cdot 10^4$, 1 min in the model corresponds to 3.8 – 38 kyr in nature.



Experimentalists are always in search of a perfect dynamic scaling for their models. Scaling all the aspects of geological
processes is very difficult to achieve, if not impossible (Reber et al., 2020). For example, using granular materials leads to a
length scaling inherently not perfect: grains in the order of $0.1 - 1$ mm in laboratory, assuming a length scaling factor of $10^{-5}$,
would correspond to 10 to 100 m in nature, which is obviously overestimated. For landscape evolution models even more
issues are linked to fluid flow or sediment transport (Paola et al., 2009). Nevertheless, these experiments provide an
"unreasonable effectiveness" (Paola et al., 2009) that allows for their interpretation in terms of scaling by similarity (Reber et
al., 2020). When the models and their natural prototype behave in a similar way, it is indeed possible to infer information about
the prototype studying the processes acting on the model (Reber et al., 2020), because of the scale independency of the
processes.

## 3 Results

We present six different models carried out with the same boundary conditions (i.e. imposed slope, precipitation rate and
experimental time) and different mixed materials (Tab. 1). The same models have been conducted multiple times to ensure the
reproducibility of the experimental results. The results presented in this paper are just a selection from amongst these
repetitions. We tested mixes with different concentration of CQ, SP, GM and PVC, accounting for the divergences in erosional
responses, starting from what is already known in literature. These materials (but CQ) are also used in Graveleau et al. (2011)
for the authors selected mix, named "MatIV", composed of 40% SP, 40% GM, 18% PVC and 2% graphite powder. The same
mix is here represented by CM2, where PVC is 20 wt.% due to the absence of graphite. Therefore, this mix has been set as our
reference model for further analysis. From this starting point, we increased the SP concentration (from 40 to 70 wt.%, from
CM2 to CM5) lowering the GM and PVC concentration. Two experiments were carried out using only SP (SM1) and a mix
with the same proportion of CM2 but with SP replaced by CQ (CM1).

The analysis illustrates the erosional properties showing the influence of different composition on the morphology of the
landscape, the river longitudinal profile, the sediment discharge and erosion. All eroded material leaves the system; therefore,
sedimentation is not modelled.

### 3.1 Morphology of erosion

In the reference experiment CM2 the rainfall system induces channel incision and triggers mass wasting processes of portion
of the analogue materials adjacent to stream channels. Both advection in channels and diffusion processes on hillslopes are
present. A well-developed river network evolves 5 minutes after initiation. Single channels coalesce in basins with the increase
of erosion and they are separated by sharp ridges. Three main basins are located at the upper part of the model (Fig. 5). The
planar surfaces developing close to the lowermost side of the experiment have a slope of about 12°, three degrees lower than
the initial imposed slope. The bottom of the valleys and the peak are generally separated by 1-2 cm relief (Fig. 6). CM1
evolution differs substantially from the reference model CM2. No channel incision is observed (Fig. 5), while diffusion and





mass wasting are the dominant processes. Two different planar surfaces are formed, separated by a vertical scarp convex upward. An elongated elevated body stands close to the left boundary, related to the boundary effect itself. The planar surfaces have a slope of 13° and 15° for the lower and upper surface, respectively. The 13° slope is reached after 5 min from the beginning of the experiment, where a proto scarp is already formed. Subsequently, the scarp moves backward for about 60 min at a rate > 0.4 cm min$^{-1}$, following which the scarp continues retreating but at a slower rate (< 0.1 cm min$^{-1}$). The difference

in elevation between the two planar surfaces is 2-3 cm. In the experiment SM1 channel incision is strong and affects all the model. Almost no mass wasting processes are observed. The landscape evolution for this model is similar to CM2. Four main basins are observed (Fig. 5), with a series of smaller basins linked to the major ones. Two of these basins stand on the leftmost part of the model and are separated by two main ridges (Fig. 6). The rightmost basins have a small ridge separating them. The ridges between different basins can attain a slope close or even higher than 90°. The planar surfaces that form at the end of the

experiment have a slope between 9° and 10°, 6° or 5° lower than the imposed initial slope. On the slopes bordering the basins several small channels form. Further increasing the SP concentration changes the erosional response of the model (Fig. 5, 6). Channel incision becomes the main process acting on the model with the SP concentration from 40 wt.% to 50 wt.%. Further increase in the amount of SP produces more and narrower channels (Fig. 5). An anastomose system develops in CM5. In CM3, CM4 and CM5 the morphologies develop after around 10 minutes from the beginning of the experiment and are almost constant

through the evolution of the models. No proper basin develops in these models and there is no evidence of diffusive processes on hillslopes. As a matter of fact, swath profiles transverse to the rivers show strong variation in elevation with a small wavelength (Fig. 6). The valleys are sharp and are very close each other and are not very incised.

### 3.2 River longitudinal profiles

The river longitudinal profile represents the variation of stream elevation relatively to the distance from the outlet. In Fig. 7

we show a river profile for each experiment at four different timesteps. The river evolution in the reference model CM2 follows a well-known path, starting from the undisturbed initial slope and arriving at the final profile with a concave-upward shape. We can also observe how the propagation of the perturbation, from the initial condition, migrates from the outlet to the headwater as a knickpoint separating the transient from the equilibrium channel profile. In the upper parts of the model, the erosion removes up to 3-3.5 cm of material. In CM1 no proper river develops and the main knickpoint defines the entire

topography, with two planar surfaces separated by a sharp scarp (Fig. 5). The experiments of CM3, CM4 and CM5 show a common behavior. Channels do not show a concave-upward shape, or maybe only in the uppermost part of the model, while generally straight rivers develop. Nevertheless, we can observe the propagation of the erosion wave from the bottom to the top. As in the other models, the incision is strong, but it does not produce very deep valleys (1.5-2 cm). Finally, in SM1 it is difficult to observe a proper concave upward river profile. Some knickpoints in the earlier stage of river development are later

obliterated (green profile in Fig. 7). The incision removes almost 3 cm in the northernmost portions of the model.



### 3.3 Sediment discharge and erosion

Sediment (mass) discharge can be characterized by the amount of material that leaves the model. Keeping the boundary conditions constant in all the experiments, its evolution is only function of the analogue material composition. Sediment discharge plotted over time shows always two main phases (Fig. 8): phase I, fast removal of material from the model; phase

II, slower removal of material with a lower discharge rate that is kept constant until the end of the experiment. After an initial period in which the material quickly responds to the boundary condition with a high discharge rate that varies through time (the slope of the solid line in Fig. 8, phase I), the system reaches an equilibrium with an almost constant discharge rate (the slope of the dashed line in Fig. 8, phase II). In the reference model CM2 and in SM1 this occurs when basins reach the dimension of 40-80 cm$^2$. Different behaviors are shown by CM3, CM4 and CM5, where the phase I is extremely short (Fig.

8). In the reference model CM2 phase I last at least 80-90 min with a discharge rate around 15 g min$^{-1}$, while for phase II is 6 g min$^{-1}$ (both values are comparable with SM1). In CM1 the discharge rate decreases from phase I to phase II, from ca. 31 g min$^{-1}$ in the first 60 min to ca. 7 g min$^{-1}$ respectively. The loss in discharge rate between phase I and phase II is around 76% and is linked in time to the stationary conditions in the morphological evolution of the experiment. SM1 shows a similar trend, but a late and smoother transition from phase I to phase II happening after 140 min from the beginning of the experiment. The

discharge rate is around 17 g min$^{-1}$ during the phase I, and 6 g min$^{-1}$ during the phase II. A strong decrease in sediment discharge over time is observed between CM2 and CM3, while from CM3 to CM5 the difference is smaller. In CM3, CM4 and CM5 phase I is very short in time (< 20 min) with a discharge rate that decrease with the increasing of SP concentration (from 19 g min$^{-1}$ to 13 g min$^{-1}$). Phase II then lasts for the rest of the experiments, with a discharge rate of about 3 g min$^{-1}$.

In Fig. 9 we show the evolution of the erosion for the selected mix. In CM2 and SM1 river channels and basins are observed.

In CM2 they are initially wider than in SM1. CM2 develops less basins, and they are less elongated. In CM2 the erosion appears to be more efficient, with a removal of material up to 3 cm in the uppermost portion of the model, close to the middle part of it. The erosion in CM1 follows the retreat of the scarps and it is mainly focused from the scarp to 5-6 cm over the outlet (Fig. 9). It wipes out at least 3 cm of the model, although no channels form. The erosion is likely homogeneous on the lower planar surface. The channel incision in CM3 is evident and it reaches 2 to 2.5 cm depth. Here the erosion is focused in the

channels, and they do not coalesce into basins. In CM4 and CM5 this affects even more the models. The erosion and incision are light reach a depth lower than 2/2.5 cm (even lower in CM5).

### 4 Discussion

The experimental setting presented in this paper allows for the investigation of the materials and composite materials response to the applied boundary conditions (15° slope and precipitation rate of 25-30 mm h$^{-1}$). Despite simplifications (i.e., lack of

tectonics, vegetation, storms, chemical weathering, seasoning, presence of infrastructures), these models highlight how the composition of the experimental material controls its erosional response. Respect to other works on the same topic (e.g. Graveleau et al., 2008, 2011), we focus on how varying the concentration of materials in a mix affect the models. Increasing



the concentration of SP respect to GM and PVC results in straighter channels and lower incision. Using CQ instead of SP results in an almost uniform morphology where no proper basins or channels form, and where the erosion is mainly due to fast

discrete events. Three main aspects arise from our results: a) SP is a key ingredient to properly model erosion, b) the physical properties of the experimental mix influence the sediment discharge rate and c) the experimental results can be used to better understand how surface processes act in nature. These considerations lay the foundations for choosing the proper analogue material for landscape evolution models. This indeed must satisfy conditions like morphology of the river channels, geomorphic indexes and erosional behavior, that should fall in the range of natural observations. We find that the best analogue

material for landscape models settled in this work is represented by the composite materials used in models CM2 and CM3 (40 wt.% SP, 40 wt.% GM, 20 wt.% PVC and 50 wt.% SP, 35 wt.% GM, 15 wt.% PVC, respectively). Even if our compositions are comparable with the ones used in other laboratories (e.g. Graveleau et al., 2011), it is important to define how different combinations of the proposed materials may result in extremely different model evolution.

## 4.1 Comparison with previous work

The study by Graveleau et al. (2011) represents the recent foundation for modelling landscape evolution. Therefore, we start discussing differences and similarities of our results with respect to their study. In Graveleau et al. (2011) the authors tested four pure materials (silica powder, glass microbeads, PVC powder and graphite) and a single composite material (named "MatIV"). The composite material is composed by 40% silica powder, 40% glass microbeads, 18% PVC powder and 2% graphite. In our study we did not analyze graphite. We tested the effect of crushed quartz in composite materials, instead.

Concerning pure materials (silica powder, glass microbeads and PVC powder), our estimations and measurements for sphericity, grain size, density and permeability match the ones made by Graveleau et al. (2011) with unavoidable minor differences. We measured higher values of porosity for PVC, while permeability is in the same order of magnitude. Internal friction angles at peak and stable friction here presented are consistently lower for all our materials in comparison with Graveleau et al. (2011). The authors settled the tests at lower normal stresses than our measurements ($< 5\,\text{kPa}$ and $< 250\,\text{kPa}$,

respectively). The Mohr-Coulomb failure criterion shows that when low normal stresses are applied to the sample, the failure envelope tends to steepen, inducing values of internal friction angle higher than if measured at higher normal stresses (Schellart, 2000). This could explain the differences in results.

The composite granular material presented by the author ("MatIV") has been proposed in this work as analogue material used in CM2, except for graphite powder replaced by a slightly higher amount of PVC powder (see Results for more details).

Density, porosity and permeability are comparable with what has been measured by Graveleau et al. (2011) for MatIV. The values for peak friction and stable friction measured in this work are comparable to what has been measured by Graveleau et al. (2011).

Erosion of the models also show similar evolution. The same rate for precipitation has been adopted in both works. We can observe strong similarity in the landscape reorganization between "MatIV" and CM2, looking at the frames in the evolution

of the models at 15° slope (Fig. 9a-e, 10c in Graveleau et al. (2011) and Fig. 5 of this manuscript). The mass discharges over





time (Fig. 9f in Graveleau et al. (2011) and Fig. 8 of this manuscript) for SilPwd (SM1 in this work) and "MatIV" coincide with our curves for SM1 and CM2 (at least for the first 90 min). The convergence of results grants the reliability of the experimental method, carried out independently at two different laboratories and by different working groups.

## 4.2 Silica powder for erosion models

SP is widely used in geomorphic experiments showing a good qualitative response to erosion/sedimentation and developing geomorphic markers that morphologically approximate the natural prototype (e.g. Bonnet and Crave, 2006; Graveleau et al., 2011; Schumm and Parker, 1973; Tejedor et al., 2017). As already stated by Graveleau et al. (2011), pure granular materials such as SP, GM or PVC are not able to fully satisfy the requirements for our analogue models. Pure GM and PVC show a good internal deformational style in convergent settings, but lack in reproducing realistic landscape morphology, while SP shows

the opposite behavior (Graveleau et al., 2011 and references therein). A weighted mixture of these three components is then needed to fulfill the requirements for a scaled analogue model, in terms of deformation and erosional style. We managed to pin down silica as main component in our composite materials, and we tested two different siliceous materials: CQ and SP. They are almost identical in their chemical composition (Fig. 2), but they strongly differ for grain size, sphericity and roughness (Fig. 1, Tab. 2). CM1 and CM2 have the same percentage of materials, but SP and CQ are switched. CM1 does not show

channel incision, while CM2 is characterized by channel incision and mass wasting processes (Fig. 5). The channel incision becomes the main process acting on the surface moving from CM2 to CM5 (from 40 to 70 wt.% of SP), but an increase in the number of channels (Fig. 5) produce less incised structures (Fig. 9). Despite 100 wt.% of SP, SM1 does not develop only straight channels, differing from CM3 and CM5. We can state that the morphological response to erosion depends on the geometrical and physical parameters rather than on the chemical ones (Fig. 5, 8, 9, 10). Indeed, all the materials do not

chemically react with each other and with water. The ratio $S_r = P/I_c$, where $P$ is the precipitation rate and $I_c$ the infiltration capacity, strongly controls the evolution of the experiments (Graveleau et al., 2011). When $S_r < 1$, $I_c$ is greater than $P$, and most of the water coming from the raining system is drained internally, inside the porous material. A flow at the interface between the model and the bottom of the box develops, triggering mass wasting processes. This configuration makes very hard to develop a well-defined surface runoff, and we could expect the same results as in CM1. On the other hand, when $S_r > 1$, the

precipitation rate allows for the developing of a river network at the model surface (as in CM2). Of course, it is possible to slightly change the precipitation rate according to the purpose of the experiment, but the main control on $S_r$ is exerted by $I_c$. This parameter is function of the permeability and, in turn, it is function of the grain-size, grain-size distribution and (effective) porosity (Carman, 1938, 1956; Kozeny, 1927). These factors are responsible for the differences between CQ and SP and are then responsible for the results of CM1 and CM2 (same concentration of materials but with CQ and SP, respectively). In fact,

the grain size of CQ is one order of magnitude higher than SP, and the latter has a higher grain size distribution (Fig. 1). Permeability spans over two order of magnitude, from $2.34 \cdot 10^{-12}$ m² to $3.56 \cdot 10^{-14}$ m² for CQ and SP, respectively. The mix CM1 show higher permeability than CM2. These consideration lead to assess SP as best siliceous material for landscape evolution models rather than CQ. But due to the very small grain size of SP, suction and capillary forces are very strong when



water is involved. Consequently, other components become necessary to promote mass wasting processes and for smoothing
the mechanical behavior of SP to avoid unrealistic brittle structures (Graveleau et al., 2011). But going beyond what has already
been done, we focused on testing how different combinations of SP, GM and PVC change the model results. Increasing the
SP concentration should bring to configurations similar to SM1, but our results show that this is not the case. When SP is ≥ 50
wt.% of the composite material, only straight channels form, and they are not so incised if compared with the ones from SM1
or CM2 (Fig. 5, 9). CM3, CM4 and CM5 do not develop basins, and the erosion in the channels is limited (Fig. 9). Mass
wasting processes and gravitational processes are absent, and the rivers flow in narrow canyons. In CM5 the behavior is even
more peculiar and anastomose channels form with a very low incision (Fig. 5, 9). Thus, a mixture of SP, GM and PVC, where
the former one has the highest concentration, develops forms that are very different from SM1, even if SP is 70 wt.%. We
propose here that this is mainly due to the grain size distribution. The voids between the grains of GM and PVC are most likely
filled with the material with the lowest grain size, SP. If the concentration of SP is high enough (≥ 50 wt.%) and water is added
to the system, the strength of the material increases, and the erosional and mechanical response of the mix strongly change.

### 4.3 Sediment discharge rate as function of physical properties of analogue materials

Phase I and phase II differ in terms of sediment discharge rate ($sdr$). Independently from the material, phase I displays higher
$sdr$ than phase II. In phase I the models equilibrate with the boundary conditions imposed. The amount of potential energy
triggers a fast reorganization of the system. In the first time steps the materials quickly leave the models, until the energy
decreases and a new equilibrium is reached. Lowering the model slope toward an equilibrium shape slows down the erosional
response of the model, entering in phase II (Fig. 8). In this latter phase the system has reached a balance with the boundary
condition, and $sdr$ shows a lower variability for a given model (Fig. 10). Despite these common points, features like the onset
of the phases, their duration or the amount of discharged material differ among the models. During phase I all the models show
a high variability in $sdr$ (Fig. 10). Nevertheless, CM1 still has the highest mean $sdr$ (red line in Fig. 10), that later decreases
from CM2 to CM5. CM1 erodes through fast and discrete processes (e.g. mass wasting), while all the other models show also
(or only) channel incision (Fig. 5). From CM2 to CM5 the $sdr$ decrease. Due to the absence of chemical reaction between
components and water, the differences in $sdr$ are linked to the physical properties of the materials. In CM1 the subsurface
water flow induces collapse of material in catastrophic events. In SM1 this does not happen. The very low permeability of the
material inhibits significant subsurface flows. Here the erosion of the material is mainly linked to the ability of water to detach
particles from the riverbed and carrying them outside the model. Initially, particles are detached when shear stresses exerted
by water overcome the threshold for detachment of the grains in the analogue materials (Howard, 1994). The strength of SP
thus controls the rate of incision. Similar considerations for CM2, but here the higher permeability can trigger both channel
incision and gravitational processes. Surprisingly, CM3, CM4 and CM5 response to precipitation rate is very different from
SM1 or CM2. From CM3 to CM5 the $sdr$ strongly decrease in both phases (Fig. 10), even if components like GM and PVC
are added to the composite materials. The properties of these two pure materials would produce an erosional response similar
to CM1 (Graveleau et al., 2011). However, when SP is in a proportion higher than 50 wt.% the water capacity of detaching





particles strongly decrease, so that even the incision is very shallow (Fig. 9) and so the *sdr*. No mass wasting processes act on these models, as suggested by the low permeability. We propose that the higher grain size distribution allows SP to fill the voids between the GM and PVC particles, lowering the permeability (Tab. 2) and increasing the material resistance with

capillarity and electrostatic forces. In phase II the *sdr* variability is smaller, and the mean values are more representative of the whole *sdr*. The previous consideration on the role of grain size applies also in this phase, even if *sdr* is significantly lower.

### 4.4 Drainage network morphology

Hack's Law (Eq. 11) can be also written as

$$L = cA^h \tag{13}$$

where $L$ is the length of the channel in a basin and $A$ its drainage area, $c$ is a scaling coefficient and $h$ is the scaling exponent referred to as Hack's exponent. The scaling coefficient $c$ and the scaling exponent $h$ are related to $k_a$ and $H$ in Eq. 11 by $c = k_a^{-1/H}$ and $h = 1/H$, respectively. Hack's Law represents the relationship between channel length and drainage area, and allows to analyze the geometry of the drainage network. Dodds and Rothman (2000) show that $h$ is in the range 0.44-0.56, while $c$ is between 1.3 and 6.6 (for individual basins compared at their outlets). Values of $h$ greater than 0.5 are typically interpreted as

relative to basin elongation with increasing size (Rigon et al., 1996). Our values for $h$ are systematically higher than 0.5 (Fig. 11), with SM1 and CM2 that show values that are lower with respect to the other models. The scaling coefficient $c$ for SM1 and CM2 is in the range 1-4 and 0-4, respectively (between the 25th and 75th percentile) (Fig. 11). CM4 shows values for $c$ close to 1, while CM5 has a slightly larger distribution. CM1 and CM3 show the lowest values for the scaling coefficient $c$. Comparing the length-area scaling of our analog models with observations (Fig. 5) we notice that the models are characterized

by a very low degree of branching of the drainage network. The drainage basins are typically elongated, especially for CM3, CM4 and CM5. SM1 and CM2 still have high values for $h$, but lower with respect to the other models. For SM1 and CM2 the basins are morphologically better defined (Fig. 5).

### 4.5 Steepness and concavity index

We must point out that our models are not meant to simulate specific landscapes, but to explore how material properties

influence landscape development. Despite the unavoidable limitations and simplifications of the model, it is tempting to compare the experimental and natural data.

We use the following metrics for quantifying erosion in both the laboratory and nature: $k_{sn}$ and $\theta$. Both $k_{sn}$ and $\theta$ represent a 1:1 metric for lab- to nature comparison. $\theta$ is dimensionless while $k_{sn}$, whose dimensions are a function of a reference concavity $\theta_{ref}$) has been computed considering the length scaling factor $h* = 10^{-5}$, so that the values for $A$ in analog models in Eq. (9) are

in m². For calculating $k_{sn}$ in analog models ($k_{sn\_MOD}$) we assumed $\theta_{ref} = 0.45$, similar to studies on natural landscapes. We analyzed river profiles of phase II of the experiments because this phase is linked to equilibrium of the system. In general, $\theta_{\_MOD}$ tends to be lower than 0.5 with the exception of CM2 and CM3, due to the straightening of river longitudinal profiles during the model run (Duvall, 2004; Whipple and Tucker, 1999). Despite the scattering of values for $\theta_{\_MOD}$, SM1, CM2 and




CM3 show average values higher than the other models, from 0.2 to 0.5 (for data between 25th and 75th percentile). For $k_{sn\_MOD}$
we found that values computed during phase II range generally between 10 and 140 m$^{0.9}$ (Fig. 12). The values for $k_{sn\_MOD}$ and
$\theta_{\_MOD}$ do not allow for a unique discrimination between the types of erosion affecting the models, in terms of detachment-
limited erosion and transport-limited erosion (Tucker and Slingerland, 1997; Tucker and Whipple, 2002; Whipple and Tucker,
2002), but we consider it likely that experiments are often transport-limited rather than detachment limited. The concavity
index for detachment-limited streams are typically higher than for transport-limited streams (Brocard and Van der Beek, 2006;
Whipple and Tucker, 2002), even if there is some evidence which suggest that this might not always be true (Gasparini, 1998;
Massong and Montgomery, 2000; Tarboton et al., 1991). Of our models, CM2 and CM3 show the highest values for $\theta_{\_MOD}$,
while CM2 and SM1 show the highest values for $k_{sn\_MOD}$. Both $k_{sn\_MOD}$ and $\theta_{\_MOD}$ (Fig. 12) are generally comparable with data
coming from natural compilations (e.g. Kirby and Whipple, 2012). The matching of $k_{sn}$ and $\theta$ between models and nature
supports future development and application of the analog materials tested in this study for modeling landscape evolution.

## 5 Conclusions

We used mixes of water-saturated granular materials as analogs for the upper brittle crust analyzing the role played by
geometrical and physical properties in landscape evolution models. Our experimental results illustrate how small variations in
the composition of an analogue material can strongly affect the evolution of the geomorphological features and the mechanical
response of the materials. According with previous works, we find in SP the main component of analogue materials for
landscape evolution models, better if mixed with GM and PVC. We can now conclude that:

a) granular materials and mixes of them deform following Mohr-Coulomb criterion. Adding GM and PVC to SP smooth
   the deforming properties of the SP, allowing for the formation of brittle structures;
b) composite materials with smaller grain size distribution and higher grain size (order of 100-200 µm) do not allow for
   advection in valleys, due to higher permeability. The sediment (mass) discharge rate is high, and the erosion happens
   quickly in time;
c) composite materials with higher grain size distribution with particles in the order of 10s of µm allow for both channel
   incision in valleys and diffusion on hillslopes;
d) composite materials where the percentage of SP higher or equal to 50 wt.% show high number of channels but with
   a very low incision. The discharge rate is extremely low, and erosion and incision affect less the model.
e) respect to the other models, SM1 and CM2 show more branching and well-defined basins, while CM2 and CM3 show
   higher values for concavity index.

The geomorphological observations carried out on the models here presented highlights how SM1, CM2, CM3, show features
most similar to natural prototypes. Increasing the SP concentration from 40 to 50 wt.% (CM2 and CM3, respectively) leads to
straighter channel better defined. For models coupling tectonics and surface processes, the material used in SM1 is not likely
to be adequate, due to its poor deformational behavior. The Hack's exponent in all models was higher than observed in nature,



but SM1 and CM2 exhibited the lowest values. Concavity index for all models tended towards values lower than in nature, except for CM2 and CM3, which showed good agreement with nature. All these considerations suggest that the materials used in models CM2 and CM3 should be implemented for reproducing analogue landscapes. Even if our findings are in agreement with previous works, here we also quantified the differences between geomorphological indexes as function of composition
of analogue materials, giving a further constrain on the choice of the materials. These mixes will be adopted in contexts of active tectonics in future works.

**Data availability.**

Digital images, topographic data from laser scans, scripts and raw data have been uploaded using GFZ Data Services and can be accessed through http://pmd.gfz-potsdam.de/panmetaworks/review/d63845bbb81e2460d1e1e69dfbb0a189719d5d4de9364314c30d814e3695d1e9/.

**Acknowledgments.**

We would like to thank Diego Sebastiani, Anita Di Giulio, Maurizio Di Biase and Andrea Di Biase from the geotechnical laboratory at Università "La Sapienza", for the fundamental help and for the useful discussions. We thank Stéphane Dominguez for providing us the materials that have been used as comparison to the materials presented in this manuscript. We also thank TPV Compound s.r.l. (Frosinone, Italy) for the helpfulness in providing us with PVC powder, and CNG Srl for their
laboratories. The Grant to Department of Science, Roma Tre University (MIUR-Italy Dipartimenti di Eccellenza, ARTICOLO 1, COMMI 314 – 337 LEGGE 232/2016) is gratefully acknowledged.

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

| Experiment | Analogue material/mix |
|---|---|
| SM1 | 100 wt.% SP |
| CM1 | 40 wt.% CQ |
|  | 40 wt.% GM |
|  | 20 wt.% PVC |
| CM2 | 40 wt.% SP |
|  | 40 wt.% GM |
|  | 20 wt.% PVC |
| CM3 | 50 wt.% SP |
|  | 35 wt.% GM |
|  | 15 wt.% PVC |
| CM4 | 60 wt.% SP |
|  | 30 wt.% GM |
|  | 10 wt.% PVC |
| CM5 | 70 wt.% SP |
|  | 25 wt.% GM |
|  | 5 wt.% PVC |

**Table 1: Material used in experiments. The label CM indicates a composite material, while the label SM indicates a pure single material.**



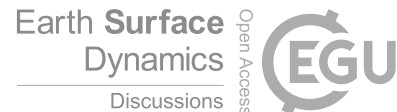

| Material | $D_{50}$, $\mu m$ | $\rho_{dry} \pm 1$ $kg\ m^{-3}$ | $\gamma \pm$ $0.01$ | $k \pm 0.01 \cdot 10^{-n}$ $m\ s^{-1}$ | $k$ $m^2$ | $\phi^p_{wet}$ (degrees) | $C_{wet}$ (kPa) | $\phi^s_{wet}$ (degrees) | Sphericity | Roundness |
|---|---|---|---|---|---|---|---|---|---|---|
| **Silica powder (SP)** | 20 | 2661 | 0.36 | $3.47 \cdot 10^{-7}$ | $3.56 \cdot 10^{-14}$ | 34-40 | 0-8.5 | 33-39 | Low | Very angular |
| **Crushed quartz (CQ)** | 87 | 2588 | 0.37 | $2.28 \cdot 10^{-5}$ | $2.34 \cdot 10^{-12}$ | 30-33 | 4.5-6.6 | 31-32 | Medium | Angular |
| **Glass microbeads (GM)** | 98 | 2452 | 0.26 | $2.80 \cdot 10^{-5}$ | $2.87 \cdot 10^{-12}$ | 23-25 | / | 14-22 | Very high | Well rounded |
| **PVC powder (PVC)** | 181 | 1402 | 0.30 | $1.03 \cdot 10^{-5}$ | $1.06 \cdot 10^{-12}$ | 25-32 | / | 18-21 | High | Rounded |
| **CM1** | / | 2172 | 0.42 | $7.23 \cdot 10^{-6}$ | $7.42 \cdot 10^{-13}$ | 25-40 | 0-8.9 | 23-35 | / | / |
| **CM2** | / | 2192 | 0.32 | $2.83 \cdot 10^{-6}$ | $2.90 \cdot 10^{-13}$ | 25-36 | 1-9.8 | 23-34 | / | / |
| **CM3** | / | 2285 | 0.29 | $9.01 \cdot 10^{-7}$ | $9.25 \cdot 10^{-14}$ | 27-40 | 0-6.5 | 26-36 | / | / |
| **CM4** | / | 2386 | 0.30 | $2.56 \cdot 10^{-6}$ | $2.63 \cdot 10^{-13}$ | 22-37 | 0-11.9 | 22-36 | / | / |
| **CM5** | / | 2496 | 0.31 | $3.96 \cdot 10^{-6}$ | $4.06 \cdot 10^{-13}$ | 22-36 | 2.9-14 | 21-35 | / | / |

**Table 2: Geometrical and physical properties of pure granular materials and mixes. Here we show values for grain size ($D_{50}$), density ($\rho_{dry}$), porosity ($\gamma$), permeability ($k$), cohesion ($C$) and internal friction angle for peak ($\phi^p_{wet}$) and stable ($\phi^s_{wet}$) friction. Sphericity and roughness have been estimated after the acquisition of SEM imaging. The subscripts dry and wet indicate whatever the tests were made with dry materials or water-saturated materials.**








**Figure 1: SEM (Scanning Electron Micrograph) pictures of the materials tested in this work. (a) crushed quartz, (b) silica powder, (c) glass microbeads, (d) PVC powder.**



**Figure 2: SEM qualitative analysis of the material composition. On the left is the BackScattered-Electron (BSE) imaging, while on the right the Energy-Dispersive Detector (EDS) spectrum. A qualitative composition is presented for a) silica powder, b) crushed quartz and c) glass microbeads. PVC powder has not been analyzed, due to its complex composition.**



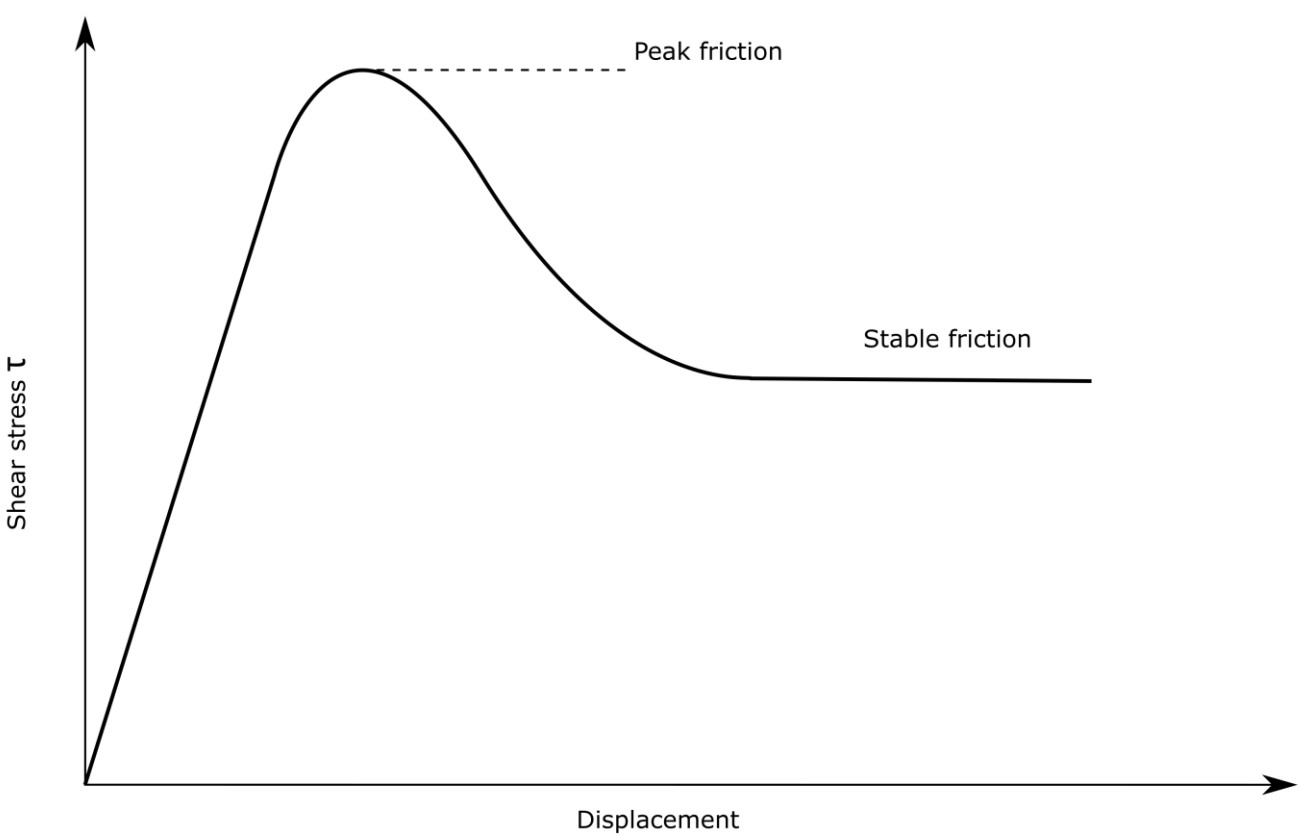

**Figure 3: Ideal output from a shear test. The peak friction corresponds to the high point of the curve, while stable friction is defined by the subsequent plateau.**




**Figure 4: Schematic representation of the experimental setup used for models with only erosion: block of experimental material (35×30×5 cm³), rainfall system (commercial sprinklers) and reclining table. A single camera and a high-definition laser scan provide records for the experiments.**



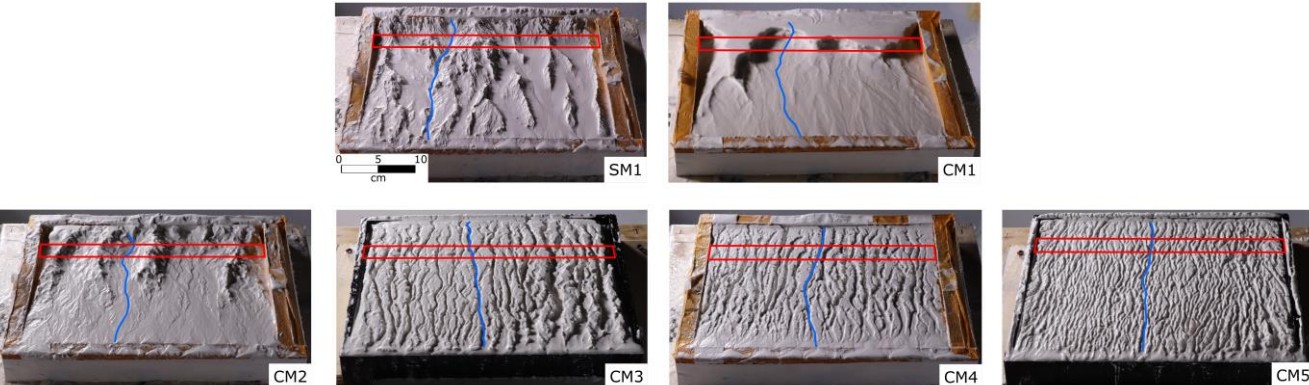

**Figure 5: Pictures of the models at 200 min from the beginning of the experiment. The red box is the trace of the swath profiles shown in Fig. 6. The blue line indicates the stream, for every experiment, analyzed in Fig. 7.**

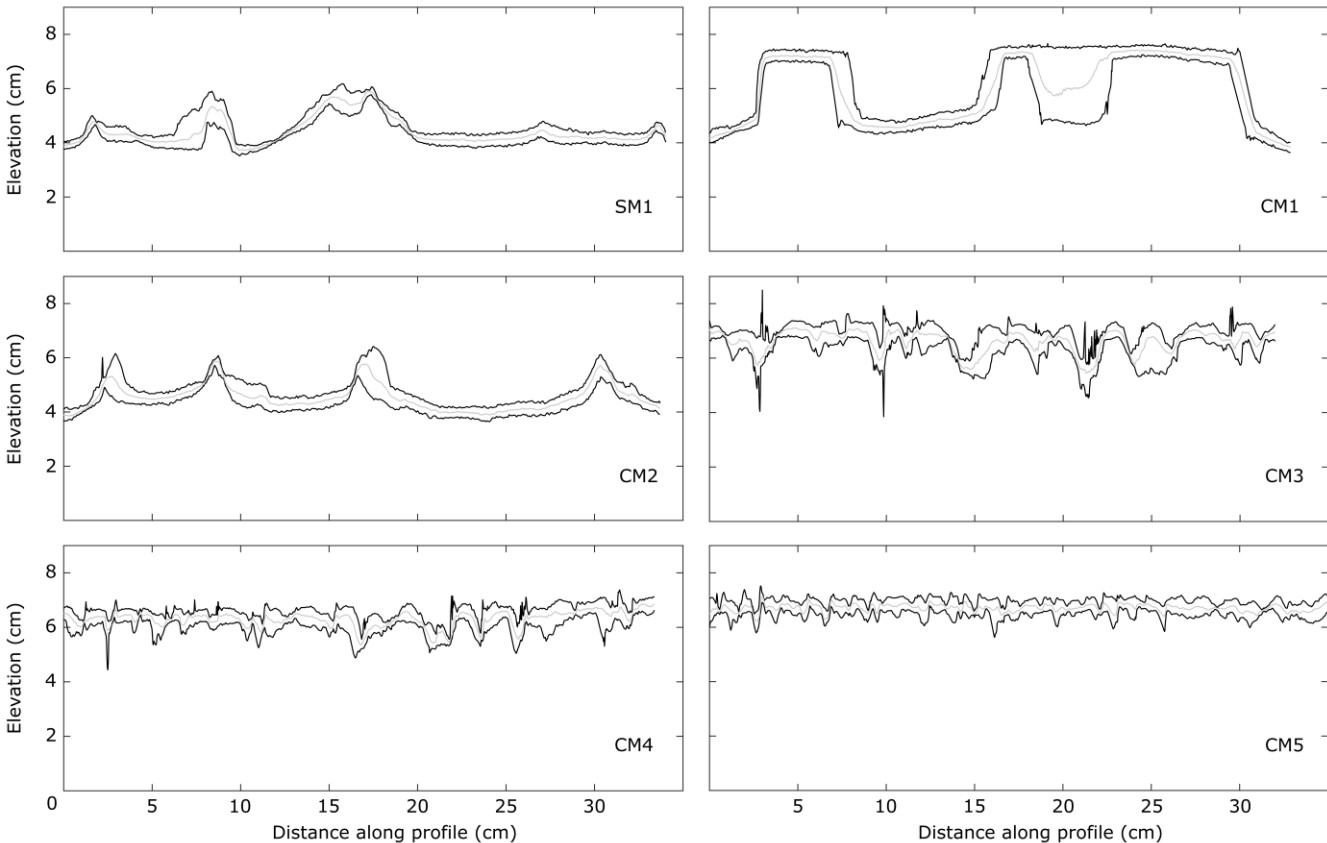

**Figure 6: Swath profiles, transverse to the experiment slope. Profiles are plotted at the same experimental time, at which the system keeps its morphologies almost constant through time (ca. 200 min). Location as shown in Fig. 5.**



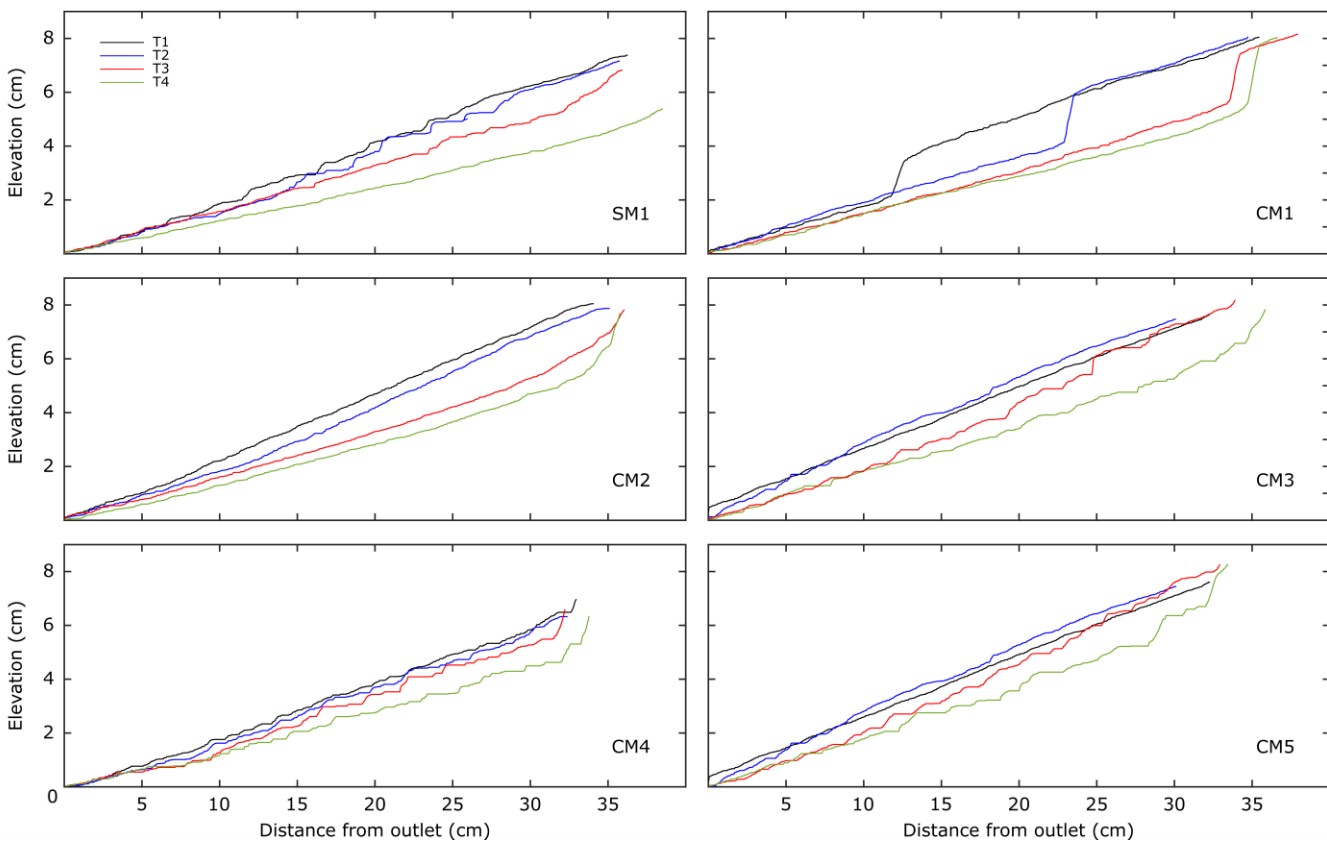


**Figure 7: Longitudinal profiles of the streams highlighted in Fig. 5. This analysis is performed converting the laser scan into DEM and applying the TopoToolbox tool for MATLAB (Schwanghart and Scherler, 2014). The laser horizontal and vertical resolution are 0.05 and 0.07 mm respectively. We show four profiles corresponding to four consecutive time steps (T1 = 20 min, T2 = 60 min, T3 = 150 min, T4 = 350 min).**



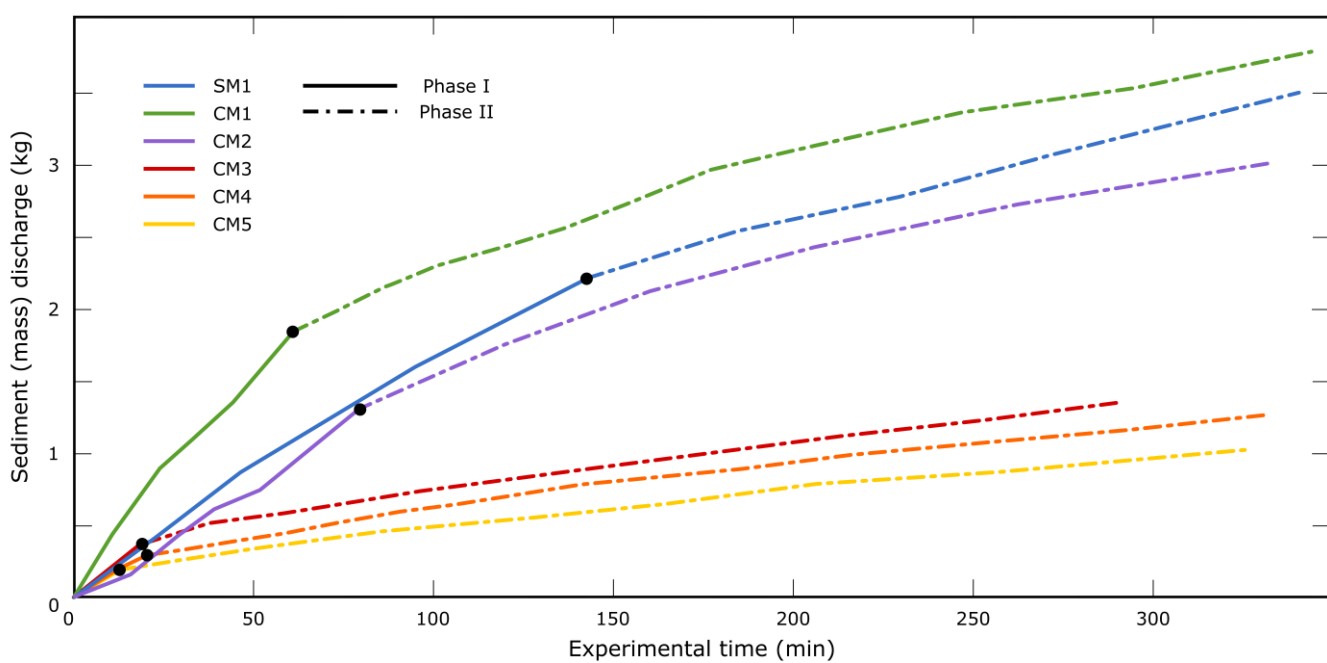


**Figure 8: Cumulative mass (sediment) discharge over time for the six experiments. The solid lines correspond to phase I, while dashed lines to phase II. The black dot highlights the transition from phase I to phase II.**







**Figure 9: Erosion evolution for the experiments, here represented by the cumulative difference in elevation ($\Delta z$) of the same point at consecutive times. Time is indicated in columns. Each row corresponds to a model, where the mix adopted is indicated in the first panel of each row. The color coding is shown by the color bar on the right. Negative values correspond to erosion, positive ones to sedimentation (almost no sedimentation for these models).**





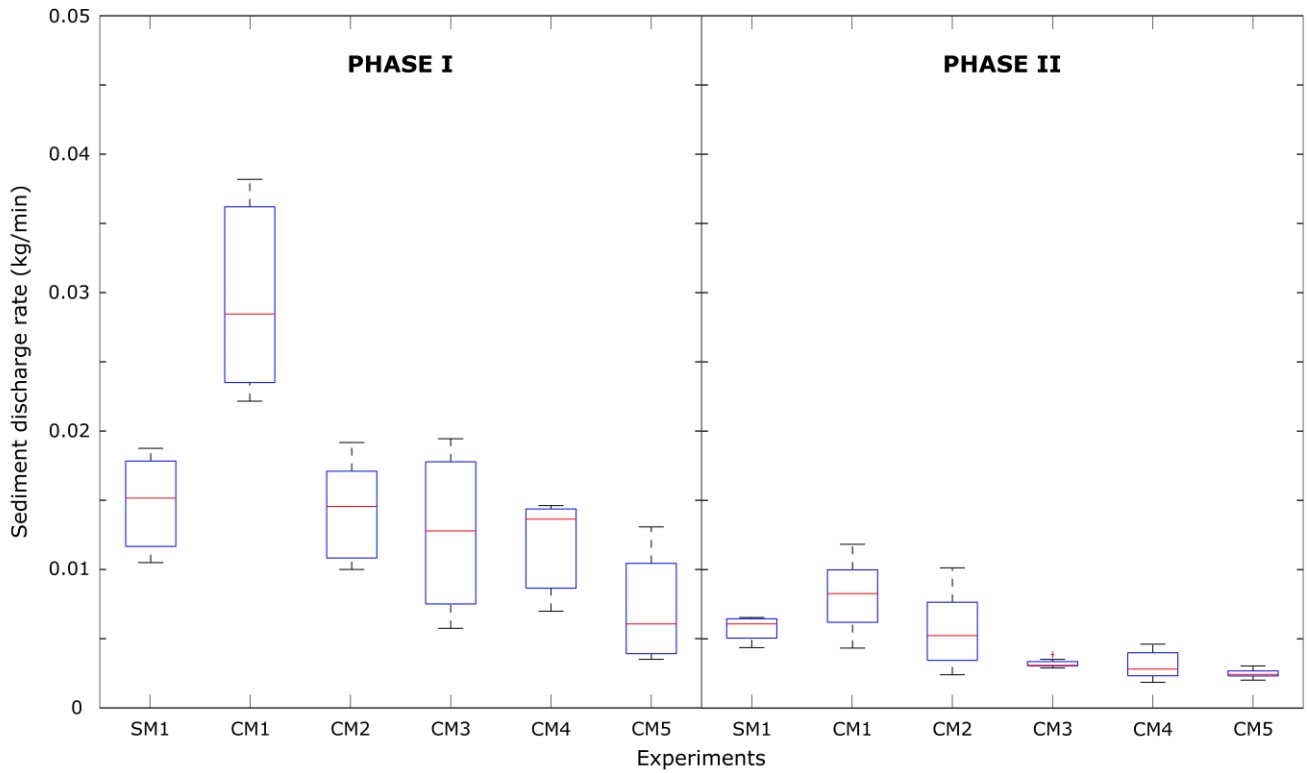

**Figure 10: Box plot of sediment discharge rate for phase I and phase II. The red lines indicate the median, and the bottom and top edges of the blue box indicate the 25th and 75th percentiles, respectively. The black markers outside the box cover the data point at < 25th and > 75th percentile that are not considered outliers.**






**Figure 11: Values distribution for *h* and *c* for Hack's Law, as given in Eq. (13), in the models. The black lines indicate the median, while the bottom and top edges of the green box indicate the 25th and 75th percentiles, respectively. The green whiskers outside the box cover the data point at < 25th and > 75th percentile that are not considered outliers, here indicated by green crosses.**





**Figure 12: Steepness ($k_{sn}$) and concavity index ($\theta$) for the experiments. We use $\theta_{ref} = 0.45$ for computing $k_{sn}$. The black dots indicate the median, while the bottom and top edges of the blue box indicate the 25th and 75th percentiles, respectively. The thin blue lines cover data <25th and >75th percentiles that are not considered outliers, and the outliers are indicated by the blue empty dots.**