# Peer review of "Erosional response of granular material in landscape models"

_Earth Surface Dynamics, 2020_

## Referee Comment (RC1) · Fabien Graveleau (Referee) · 17 Jul 2020

Review of Manuscript entitled "Erosional response of granular material in landscape models" by R. Reintano et al. The manuscript from R. Reintano et al., exposes an experimental modelling approach that aims at investigating the influence of material composition on the style of landforms (mostly drainage systems) when the material is submitted to rainfall erosion. This works goes a step beyond than an early work (Graveleau et al., 2011) that first published a mixture of granular materials suitable to be deformed under a compressive device (and actually, also in extensive and strike-slip settings) and eroded below an artificial rainfall device. In their manuscript, Reintano et al., do no deform their experimental materials, but they carefully analyzed the patterns of erosional landforms in their set of experiments. Technically, their device reproduces

roughly a classical boundary conditions in terms of slope (15°) and precipitation rate (25-30 mm/h), but a significant improvement is the accurate topographic monitoring of the model evolution, and the use of such topographic dataset in terms of erosion law parameters. In total, the authors tested 6 different materials : 1 is a raw material (i.e., 100% Silica Powder), 4 are mixtures of Silica Powder + Glass microbeads and PVC, and a last one tested crushed quartz. The results from Reintano et al.'s concern first the characterization of the properties of the different components. It is basically the geometrical properties of grains, their chemical composition, their frictional properties (both at peak and stable friction conditions), porosity and permeability. All these properties have been measured with robust devices and repeated several times. Results are in good agreement with already published values, which allows to address later in the discussion the origin of erosional behaviors observed for each materials. Then concerning model results, the experimental device designed by the authors allows to carefully explore the morphometry and erosion laws observed for their experimental landforms and to investigate the control played by material composition. Particularly, DEM analysis allowed to confront experimental results to Hack's and Flint's law, together with quantifying precisely sediment outfluxes. Erosion law parameters are quantified and compared to nature. In addition, through the analysis of dimensionless parameters (for time notably) extracted from erosion laws, this work provides a novel quantitative understanding to address scaling issues and discussions about the similarity of experimental landforms to natural counterparts. Time scaling is particularly investigated. It is addressed in another way than in Graveleau et al. (2011) publication but provides similar values. As a whole, the results obtained in this manuscript confirms that Silica Powder is a central component required to develop highly interesting and powerful experimental landforms. Several mixture composition appear suitable to develop morphotectonic experiments. In conclusion, this manuscript is a real valuable contribution that allow to goes a step forward in the demonstration that morphotectonic experiments should be considered as a useful tool to investigate active tectonics and relief dynamics questions. My comments and suggestions of corrections are really minor (see below). That's why I consider that this work definitely deserves publication at EGU Earth Surface Dynamics journal and ask for minor revision. With my best regards and all my congratulations to the authors,

Fabien Graveleau Université de Lille

Below are detailed comments on each section of the manuscript. Suggestions are written in blue and some corrections appear in red.

TITLE : OK, it fits well the content of the paper.

ABSTRACT OK, it summarize well the content of the research.

1 - INTRODUCTION L50-51 : Add one reference about S. Ouchi's work, for instance Geomorphology 2011 or 2015 ; Ouchi, S. (2011). "Developement of experimental landforms by rainfall erosion and uplift." The Journal of the Geological Society of Japan 117(3): 163-171. Ouchi, S. (2015). "Experimental landform development by rainfall erosion with uplift at various rates." Geomorphology 238(Supplement C): 68-77.

L52 : Add references by Guerit et al., 2016, Guerit, L., S. Dominguez, J. Malavieille and S. Castelltort (2016). "Deformation of an experimental drainage network in oblique collision." Tectonophysics 693, Part B: 210-222.

L57 : Cite here the reference of Bonnet & Crave (2006) since they tested (Fig.2) the effect of different mixtures of silica powder and glass beads on morphology.

2 – EXPERIMENTAL APPROACH L73 : I would cite here Lohrmann et al., JSG, 2003. Lohrmann, J., N. Kukowski, J. Adam and O. Oncken (2003). "The impact of analogue material properties on the geometry, kinematics, and dynamics of convergent sand wedges." Journal of Structural Geology 25(10): 1691-1711.

P4 Table 2: Regarding the accuracy of measurements, I am always questioned about the usefulness (and significance) to provide values with what appears to me as (over)accurate. For instance, is this really significant to provide density measurement at a 1kg/m3 accuracy ? I know that devices like the one used in this work (helium pycnometer) can account for such accuracy, but is it that significant for granular materials for which handling techniques is so important in controlling the grain packing and therefore density ? To me, I would limit the accuracy of density to 10kg/m3. This is the same for the accuracy of frictional properties. In the literature, ring shear apparatus are able to provide really accurate estimations of cohesion and frictional angle. Is it worth since granular material are sensitive to handling techniques and compaction ? In the manuscript, concerning frictional values, the authors remain in the order of magnitude of 0.1 kPa for cohesion, which is fine for me.

Density seems to correspond to "particle" (or specify) density ; that is the density of grain. So why specifying "dry" in table 1 ? It is also surprising to have such density values of grains for mixtures. Wouln't be worth to document the "apparent" or "bulk" density of the whole material ? That is the density of the material including grain and air.

L124 : Eq. 7 is quoted but it might be Eq.1.

L127 : "In this work the normal stress applied are in the range 25-200 kPa". This is normal stress applied in the Casagrande direct shear box. What about the range of normal stress in the erosional experiments ? Since the box is 5 cm deep, it should be in a range lower than the range of values applied by the Casagrande box. Would it be possible to add a few comment on this point ?

Note : In Graveleau et al (2011) we did test in a Casagrande Direct shear box but we finally decided to disregard this apparatus (and build a proper one) because the range of normal stress Casagrande box requires / could applied to the tested material was largely over the range of normal stress actually acting inside the morpho-tectonic box. As pointed later by Reintano et al., it is true that it entails that the failure envelope is steeper at such low normal stress, but we presume that it was more in the range of what is actually occurring in our models.

L134 : "ïĄľ" is mentioned in the text but it is "ïĄĘ" is Table 2. Please harmonize.

L139 : "while the mechanical properties seem to show a common trend." What do you mean by "mechanical properties" ? Not clear.

How many tests have been made per materials to obtain the average frictional properties ?

L141 : Is porosity calculated for dry materials of water-saturated materials ? I presume it is dry. . . Because they are not the same between dry and wet conditions and differences can be slight or large depending on materials (Graveleau, PhD, 2008 ; p298). This is notably the case for Silica powder where compaction is apparently more efficient when the material is wet than dry (it is at least what we observed for the Silica Powder used in the device in Montpellier).

L148 : Value of porosity for GM (0.26) is remarkably low compared for instance with Graveleau et al. (2011) who obtained 0.36, although D50 are not that different (88 $\mu$m for Graveleau et al., 98 $\mu$m for Reintano et al.). Any explanation of this ?

L157 : "The permeability values for mixes are then in the order of 10-13 m2." The averaging comment is appreciated, because, as a recall to what has been written above, proposing (in table 2) values of permeability with a 0.01 accuracy might be excessive. To me, the order of magnitude need to be conserved ; at least, 0.1 accuracy should be enough.

L173 : Eq. 10 is quoted but it might be Eq. 4. L175 : Eq. 10 and 11 are quoted but it might be Eq. 4 and 5.

L181-182 : "For the erosional behavior of the composite material, the ratio between precipitation rate and infiltration capacity appears to be the main factor controlling the geomorphological response." Yes, definitely. This recalls me a figure in my PhD (Fig.V.I p.435). It is good to mention this point and explain it here.

L214 : "Erosion and sediment discharge are computed with ad-hoc MATLAB algorithms." Any reference ?

L230-248 : I have much appreciated this temporal scaling analysis. It would be worth to discuss and compare a bit the results obtained here analytically by the authors (1 min ⇔ 3 800 – 38 000 years) to what Graveleau et al. (2011) published (1s ⇔ 100-300 years so 1 min ⇔ 6 000 – 18 000 years) in a tectonically quiescent context or others (Strak et al., 2011) published in a tectonically active extensional setting (1s ⇔ 65-375 years so 1 min ⇔ 3 900 – 22 500 years). Which is remarkably in the same range !

3 – RESULTS

L276-277 : "The planar surfaces developing close to the lowermost side of the experiment have a slope of about 12°" This angle could be rattached to a threshold angle for detachment as Lague et al. (2003) or Graveleau et al (2011) mentioned. Comments could be welcome. Compare to Graveleau et al.'s values for MatIV (8°) which is significantly lower. Why ?

L281-282 : "The planar surfaces have a slope of 13° and 15° for the lower and upper surface, respectively". Idem. Comments for the lower slope. Suggestion for the reasons why it is higher than form CM2 ?

L289-290 : "The planar surfaces that form at the end of the experiment have a slope between 9° and 10°,..." For SM1, this value of the lower angle is the same as Graveleau et al. (2011- for SilPwd (10-10.3°). So OK.

L303 : "As a knickpoint separating..." Please, locate knickpoints by an arrow on Fig 7 to be sure to look at the good place along the profile.

L307-308 : "we can observe the propagation of the erosion wave from the bottom to the top." Please locate the "erosion wave" on the figure.

L313-315 : "Sediment discharge plotted over time shows always two main phases (Fig. 8): phase I, fast removal of material from the model; phase II, slower removal of material with a lower discharge rate that is kept constant until the end of the experiment."

A parallel with Fig10 in Graveleau et al (2011) obtained differently (by weighing output sediments) can be made. The shapes of the curves are the same. It is interesting to observe that the same shape is obtained with two different techniques.

L320-321 : " while phase II is 6 g.min-1. . ." Surprisingly, it is about twice larger than the value Graveleau et al. (2011) obtained (2.8g/min) for MatIV at a 15° slope (cf figure 10 in their paper). Any idea why it is twice higher ? Maybe this comment could come in the discussion (about L375-379)

L323 : ". . .is linked in time to the stationary conditions in the morphological evolution of the experiment." This sentence is not totally clear. Please reformulate to be sure the audience will understand.

L327 " . . . that decreases. . ."

L335-336 : "The erosion and incision are light reach a depth lower than 2/2.5 cm." Sentence not clear.

DISCUSSION

L341 : "Respect to other works on the same topic . . ." Add reference from Bonnet & Crave, 2006.

L364-367 : "The authors settled the tests at lower normal stresses than our measurements (< 5 kPa and < 250 kPa, respectively). The Mohr-Coulomb failure criterion shows that when low normal stresses are applied to the sample, the failure envelope tends to steepen, inducing values of internal friction angle higher than if measured at higher normal stresses (Schellart, 2000). This could explain the differences in results." Yes, I agree. As mentioned above, a short comment on the reason why the authors choose to measure the frictional properties under normal stress conditions that might be higher than in their model could be welcome.

L400 : ". . . for the development. . ."

L420: "and the erosional and mechanical response of the mix strongly change." Which is in agreement with measurements of frictional properties. And cite Table 2.

L431 : ". . . the sdr decreases."

L441 : ". . . the sdr strongly decreases in both phases" L435-446 : A comment on the magnitude of the lower slope of the model in terms of erosion threshold and tentatively in terms critical shield stress might be worth here. I think the authors have the expertise to likely propose something. This could be related to the mechanical strength of the materials the authors measured.s

L448 : Eq. 11 is quoted but it might be Eq. 5 (or 13 ?).

L451 : Eq. 11 is quoted but it might be Eq. 13.

L455 : "Our value for h are. . ." "In our models, calculation of h are . . ."

L456 : ". . .with SM1 and CM2 that show values that are lower with respect to the other models." ". . .with SM1 and CM2 showing values that are lower with respect to the other models."

L452 – 462 : This paragraph is really interesting and worth in the publication but present phrasing renders the understanding not easy. I would suggest the authors to rephrase this section to be more straightforward. BTW, it is surprising that CM1 and CM3/CM4 have similar value of "c" but very different shape of basin.

L454-455 : "Values of h greater than 0.5 are typically interpreted as relative to basin elongation with increasing size". What about "c" ?

L464-466 : "We must point out that our models are not meant to simulate specific landscapes, but to explore how material properties influence landscape development. Despite the unavoidable limitations and simplifications of the model, it is tempting to compare the experimental and natural data". This sentence should appear earlier in the text, typically at the beginning of 4.4 section since it concerns also this paragraph

on Hack's law.

L411-472 : Figure 12 could be cited here.

L482-484 : "Both ksn_MOD and $\theta$_MOD (Fig. 12) are generally comparable with data coming from natural compilations (e.g. Kirby and Whipple, 2012). The matching of ksn and $\theta$ between models and nature supports future development and application of the analog materials tested in this study for modeling landscape evolution." What is written here is of course really important and should be strengthened to become more convincing for the audience. Presently, this comment is certainly "too short" to convince. Values of ksn or $\theta$ could be quoted for instance. The geological contexts of selected natural examples could be also mentionned.

CONCLUSION Conclusion is OK but :

L491-492 : "granular materials and mixes of them deform following Mohr-Coulomb criterion. Adding GM and PVC to SP smooth the deforming properties of the SP, allowing for the formation of brittle structures;" I don't think this could be said in the conclusion since deformation has not been tested in the paper. In addition, SP also deforms brittlely (see fig 7.c in Graveleau et al., 2011). It is more the bulk strength of SP and the tectonic style (intense fracturing) that prevent SP to be a good analogue for both morphology and deformation.

L508 : "Even if our findings. . ." I would not reduce the impact of this contribution (mostly at the end of the conclusion) but simply said that this publication goe a step forward in characterizing the erosion properties (and the origin of difference) for mixture of granular materials. I would remove "Even if".

REFERENCE I did not checked the accordance between the text and the reference list because I thrust the editorial board tools to do it properly and efficiently.

FIGURES

Figure 7 : For CM3 and CM5, it is surprising to see the T2 profile above the T1 profile.

Figure 8 : Data points should be indicated along the curves.

Figure 10 : Phase I and II could be merged on a single diagram to emphasize the decrease in sediment discharge rate between the two phases.

Please also note the supplement to this comment:
https://esurf.copernicus.org/preprints/esurf-2020-35/esurf-2020-35-RC1-supplement.pdf

---

## Referee Comment (RC2) · Michele Cooke (Referee) · 19 Aug 2020

The paper presents a nice parametric study for a range of potential analog materials that could be used in coupled deformation and erosion models. The careful study very nicely contributes to our knowledge of these materials and will be a very valuable resource for future studies. Delineating a material recipe that can effectively simulate both deformational and erosional processes within the same experiment will enable many future investigations into the fascinating coupled feedbacks between these processes. I have a few suggestions that may strengthen the paper. These suggestions primarily relate to the framing and presentation of the findings rather than the results themselves, which are nicely gathered.

[Figure]

The need for analog models of erosion presented within the introduction can be strengthened. A reader unfamiliar with analog approaches might not be convinced that these approaches are needed from reading the somewhat vague statement in the manuscript that computation capacities are limited (line 40). Can you provide some examples of the limits of computational models? One approach may be to follow the reasoning presented in Reber et al (2020) for the benefit of analog models over numerical approaches. But the authors may have other even more compelling reasons to offer the reader.

The study can benefit from stronger support for the performance assessment. The manuscript frames its primary goal as "finding an analogue material that best mimics the erosional behavior of the natural prototype." The part that is missing is how specifically is good behavior assessed. How 'should' a slope of 15 degrees respond to the simulated precipitation rate? How do we know what experimental response is correct or wrong? The text states that a proper concave upward river profile is desirable (line 309) but are there some plausible conditions that would yield more or less concavity to the profile? My, albeit very basic, understanding of fluvial mechanics is that the concavity of river profiles relates to the different strength of channel bed materials along the profile. If the material is uniform along the river profile, what degree of concavity is expected? This information is very important for helping us assess the performance of the material. Can you add to the figure 7 the range of expected profile shapes? Could you use numerical solutions for similar slope and precipitation but at crustal scale and with a range of soil/rock properties to validate the performance of the analog materials? The comparison of the laboratory observations to numerical models may be more straight-forward than comparison to natural systems because the numerical models can have uniform properties and starting slope.

Presumably, each of the tested combinations of analog materials may be suitable for different conditions of geologic substrate or precipitation rate. Perhaps one recipe is best suited for 'typical' settings but others could be used if you want to look at some

atypical geomorphic settings. Providing some guidelines for the conditions under which each recipe could be suitable (or not) would be helpful.

The comparison in section 4.5 of geomorphic parameters to measurements from crustal geomorphic systems and the limits for various processes is very interesting. Can you please add the values of K and theta for detachment limited and transport limited systems to figure 12? This will aid in the comparison. The last sentence of this paragraph asserts the match of the parameters with nature. Please add the ranges for geomorphic estimates of K and theta to figure 12.

Just a small note on porosity. If I understand correctly, the porosity is measured my comparing the volume prior to and after shaking. Doesn't this presume that they packing has no voids after shaking? If so, this doesn't seem realistic within granular materials which cannot pack with zero porosity. Maybe I'm missing something about this particular method of porosity measurement.

Specific notes: Some awkward grammar at the following lines: 19; 24-25 (rewrite 'wipe it out' as this is a bit colloquial for what you are describing); 60 (replace this argument with what you refer to); 83 (where active tectonic is present); 138-139; 145; 183-184 (inner part?); 202; 262; (divergences?); 336; 398-399; 442. Line 82: What does bivalent mean? Line 93: Like -> such as Line 247: ration -> ratio Line 264: but -> except for Line 320: comma after CM2, Line 334: Wipes out is a bit overly colloquial. 'Erodes' may be better. 350-351: This information may be more effectively conveyed within a table. 384-386: To improve clarity please expand what specifically you mean by the 'opposite behavior'? Also please explain what you mean by 'internal deformation style in convergent settings'. Do you mean the development of thrust faults? Line 410: Explain what the unrealistic brittle structures are. Line 469: Please explain that this length scaling is constrained by the strength of the granular material. Line 505: Provide specific for the poorness of behavior can be helpful.

-Michele Cooke Amherst, MA USA

**ESurfD**

Interactive
comment

---

## Editor Comment (EC1) · Jean Braun (Editor) · 15 Sep 2020

Many thanks for the thorough response to the two reviewers' comments and suggestions. I, therefore, invite you to submit a revised version of your manuscript.
* * *

---

## Author Comment (AC1) · 15 Sep 2020

We would like to thank Dr. Fabien Graveleau for the corrections and the useful comments. They significantly improved our work. We include hereafter (as supplement file), point by point, the reply (in dark red italic text) to all the Referee comments. We will also upload the revised version of our manuscript. We hope the manuscript is now ready for the publication in Earth Surface Dynamics.

Please also note the supplement to this comment:
https://esurf.copernicus.org/preprints/esurf-2020-35/esurf-2020-35-AC1-supplement.pdf

[Figure]

[Figure]

**Supplement:**

[Figure]

**Riccardo Reitano**
**Univ. Roma Tre**

To: Dr. Fabien Graveleau
Referee, *Earth Surface Dynamics*

Reference: Response to referee comments to the manuscript *" Erosional response of granular material in landscape models"* [Paper esurf-2020-35] by R. Reitano, C. Faccenna, F. Funiciello, F. Corbi, S. D. Willett.

Dear Dr. Fabien Graveleau,

we would like to thank you for the corrections and the useful comments. We include hereafter, point by point, the reply (in *dark red italic* text) to all the Referee comments. We also upload the revised version of our manuscript. We hope the manuscript is now ready for the publication in *Earth Surface Dynamics*.

With my kindest regards,

Riccardo Reitano, on behalf of all authors.

**Referee – Fabien Graveleau**

The manuscript from R. Reintano et al., exposes an experimental modelling approach that aims at investigating the influence of material composition on the style of landforms (mostly drainage systems) when the material is submitted to rainfall erosion. This works goes a step beyond than an early work (Graveleau et al., 2011) that first published a mixture of granular materials suitable to be deformed under a compressive device (and actually, also in extensive and strike-slip settings) and eroded below an artificial rainfall device. In their manuscript, Reintano et al., do no deform their experimental materials, but they carefully analyzed the patterns of erosional landforms in their set of experiments. Technically, their device reproduces roughly a classical boundary conditions in terms of slope (15°) and precipitation rate (25-30 mm/h), but a significant improvement is the accurate topographic monitoring of the model evolution, and the use of such topographic dataset in terms of erosion law parameters. In total, the authors tested 6 different materials: 1 is a raw material (i.e., 100% Silica Powder), 4 are mixtures of Silica Powder + Glass microbeads and PVC, and a last one tested crushed quartz. The results from Reintano et al.'s concern first the characterization of the properties of the different components. It is basically the geometrical properties of grains, their chemical composition, their frictional properties (both at peak and stable friction conditions), porosity and permeability. All these properties have been measured with robust devices and repeated several times. Results are in good agreement with already published values, which allows to address later in the discussion the origin of erosional behaviors observed for each materials. Then concerning model results, the experimental device designed by the authors allows to carefully explore the morphometry and erosion laws observed for their experimental landforms and to investigate the control played by material composition. Particularly, DEM analysis allowed to confront experimental results to Hack's and Flint's law, together with quantifying precisely sediment outfluxes. Erosion law parameters are quantified and compared to nature. In addition, through the analysis of dimensionless parameters (for time notably) extracted from erosion laws, this work provides a novel quantitative understanding to address scaling issues and discussions about the similarity of experimental landforms to natural counterparts. Time scaling is particularly investigated. It is addressed in another way than in Graveleau et al. (2011) publication but provides similar values. As a whole, the results obtained in this manuscript confirms that Silica Powder is a central component required to develop highly interesting and powerful experimental landforms. Several mixture composition appear suitable to develop morphotectonic experiments. In conclusion, this

manuscript is a real valuable contribution that allow to goes a step forward in the demonstration that morphotectonic experiments should be considered as a useful tool to investigate active tectonics and relief dynamics questions. My comments and suggestions of corrections are really minor (see below). That's why I consider that this work definitely deserves publication at EGU Earth Surface Dynamics journal and ask for minor revision. With my best regards and all my congratulations to the authors,
Fabien Graveleau
Université de Lille

**Main Comments**

1) P4 Table 2: Regarding the accuracy of measurements, I am always questioned about the usefulness (and significance) to provide values with what appears to me as (over)accurate. For instance, is this really significant to provide density measurement at a 1kg/m3 accuracy ? I know that devices like the one used in this work (helium pycnometer) can account for such accuracy, but is it that significant for granular materials for which handling techniques is so important in controlling the grain packing and therefore density ? To me, I would limit the accuracy of density to 10kg/m3. This is the same for the accuracy of frictional properties. In the literature, ring shear apparatus are able to provide really accurate estimations of cohesion and frictional angle. Is it worth since granular material are sensitive to handling techniques and compaction ? In the manuscript, concerning frictional values, the authors remain in the order of magnitude of 0.1 kPa for cohesion, which is fine for me. .

Density seems to correspond to "particle" (or specify) density ; that is the density of grain. So why specifying "dry" in table 1 ? It is also surprising to have such density values of grains for mixtures. Wouln't be worth to document the "apparent" or "bulk" density of the whole material ? That is the density of the material including grain and air.

*We modify the text in agreement with what has been proposed by the Reviewer. Now the measurement for density and permeability in Table 2 show an accuracy of 10 kg/m$^3$ and 0.1 m s$^{-1}$ or m$^2$, respectively.*

*As specified by the author, the density measurements here proposed refer to particle density (the name is now corrected in Table 2). As it is extremely difficult to control the amount of water inside the models during the runs, we decided to show only the particle density.*

2) L127 : "In this work the normal stress applied are in the range 25-200 kPa".

This is normal stress applied in the Casagrande direct shear box. What about the range of normal stress in the erosional experiments ? Since the box is 5 cm deep, it should be in a range lower than the range of values applied by the Casagrande box. Would it be possible to add a few comment on this point ?

*We agree with the Reviewer's comment. In the revised manuscript (lines 129-131) we have clarified this point.*

3) L139 : "while the mechanical properties seem to show a common trend."

What do you mean by "mechanical properties" ? Not clear.

*We agree with the Reviewer's comment. Since we better specified this concept in sections 4.2 and 4.3 of the manuscript, we removed this sentence.*

4) How many tests have been made per materials to obtain the average frictional properties?

*Four tests per material have been made. We added this information in the revised version of the text (line 132).*

5) L141 : Is porosity calculated for dry materials of water-saturated materials ? I presume it is dry… Because they are not the same between dry and wet conditions and differences can be slight or large depending on materials (Graveleau, PhD, 2008 ; p298). This is notably the case for Silica powder where

compaction is apparently more efficient when the material is wet than dry (it is at least what we observed for the Silica Powder used in the device in Montpellier).

*The porosity has been calculated in both dry and wet conditions. This information is highlighted in the revised version of the manuscript (lines 152-154).*

6) Value of porosity for GM (0.26) is remarkably low compared for instance with Graveleau et al. (2011) who obtained 0.36, although D50 are not that different (88 µm for Graveleau et al., 98 µm for Reintano et al.). Any explanation of this?

*We think that the reasons why the values are different may lie in the handling technique, due to the sensibility of porosity values to this. Because even if the approach for measuring porosity is different respect to Graveleau et al. (2011), we also compared our values with the ones obtained measuring the weight of the same volume of material water-saturated and after drying it in a air oven. This last approach is the same used in Graveleau et al. (2011).*

7) L214 : "Erosion and sediment discharge are computed with ad-hoc MATLAB

algorithms."Any reference ?

*Following the Reviewer's suggestion, we added in the revised version of text (lines 221-222) a part in which we suggest looking into the repository (https://dataservices.gfz-potsdam.de/panmetaworks/review/d63845bbb81e2460d1e1e69dfbb0a189719d5d4de9364314c30d814e3695d1e9/) for any code has been specifically written for this paper.*

8) L230-248 : I have much appreciated this temporal scaling analysis. It would be worth to discuss and compare a bit the results obtained here analytically by the authors (1 min → 3 800 – 38 000 years) to what Graveleau et al. (2011) published (1s → 100-300 years so 1 min → 6 000 – 18 000 years) in a tectonically quiescent context or others (Strak et al., 2011) published in a tectonically active extensional setting (1s → 65-375 years so 1 min → 3 900 – 22 500 years). Which is remarkably in the same range !

*We agree with the Reviewer's comment. In the revised version of the manuscript (lines 389-396) we have compared our results with Graveleau et al. (2011) and Strak et al. (2011).*

9) L276-277 : "The planar surfaces developing close to the lowermost side of the experiment have a slope of about 12°"

This angle could be rattached to a threshold angle for detachment as Lague et al. (2003) or Graveleau et al (2011) mentioned. Comments could be welcome. Compare to Graveleau et al.'s values for MatIV (8°) which is significantly lower. Why ?

L281-282 : "The planar surfaces have a slope of 13° and 15° for the lower and upper surface, respectively".

Idem. Comments for the lower slope. Suggestion for the reasons why it is higher than form CM2 ?

*We agree with the Reviewer's comment. We have added a discussion about the erosion threshold in section 4.3 (lines 469-481). As far as the angle for CM2 and its comparison with MatIV are concerned, we performed a better estimation of the surface slope using MATLAB, correcting the previous results. The results now show comparable values (line 286).*

10) L303 : "As a knickpoint separating…"

Please, locate knickpoints by an arrow on Fig 7 to be sure to look at the good place along the profile.

L307-308 : "we can observe the propagation of the erosion wave from the bottom to the top."

Please locate the "erosion wave" on the figure.

*We thank the Reviewer for pointing out this missing information in Fig. 7. The figure is now revised according to the Reviewer's suggestion.*

11) L313-315 : "Sediment discharge plotted over time shows always two main phases (Fig. 8): phase I, fast removal of material from the model; phase II, slower removal of material with a lower discharge rate that is kept constant until the end of the experiment."

A parallel with Fig10 in Graveleau et al (2011) obtained differently (by weighing output sediments) can be made. The shapes of the curves are the same. It is interesting to observe that the same shape is obtained with two different techniques.

*We agree with the Reviewer's comment. Indeed, that comparison has been already made in section 4.1. We have added a few lines also in the revised version of this section to better show the comparison (lines 386-388).*

12) "L320-321 : " while phase II is 6 g.min-1..."

Surprisingly, it is about twice larger than the value Graveleau et al. (2011) obtained (2.8g/min) for MatIV at a 15° slope (cf figure 10 in their paper). Any idea why it is twice higher ? Maybe this comment could come in the discussion (about L375-379).

*We are not sure about the reason why such a difference. We can speculate that the differences are mainly due to the diverse experimental approaches (e.g., handling technique, apparatus, materials etc.) adopted in the two works. However, only ad hoc tests could confirm this idea. Hence, due to the uncertainties of the case, we decided to avoid commenting this specific point.*

13) L364-367 : "The authors settled the tests at lower normal stresses than our measurements (< 5 kPa and < 250 kPa, respectively). The Mohr-Coulomb failure criterion shows that when low normal stresses are applied to the sample, the failure envelope tends to steepen, inducing values of internal friction angle higher than if measured at higher normal stresses (Schellart, 2000). This could explain the differences in results."

Yes, I agree. As mentioned above, a short comment on the reason why the authors choose to measure the frictional properties under normal stress conditions that might be higher than in their model could be welcome.

*We agree with the Reviewer, so we added a comment about the reasons behind the normal stresses adopted in the work (lines 129-131).*

14) L435-446 : A comment on the magnitude of the lower slope of the model in terms of erosion threshold and tentatively in terms critical shield stress might be worth here. I think the authors have the expertise to likely propose something. This could be related to the mechanical strength of the materials the authors measured.

*In agreement with the reviewer's comment, we add a comment about the erosion threshold in section 4.3 (lines 469-481).*

15) L452 – 462 : This paragraph is really interesting and worth in the publication but present phrasing renders the understanding not easy. I would suggest the authors to rephrase this section to be more straightforward. BTW, it is surprising that CM1 and CM3/CM4 have similar value of "c" but very different shape of basin.

*In agreement with the Reviewer's suggestion, we provided the needed changes to this section, which now includes: 1) the meaning of Hack's Law and related constants; 2) ranges in the lab and nature; 3) the*

*meaning of values of "h" larger than 0.5. We think that now the readers can better understand the meaning of the analyzed parameters.*

16) L454-455 : "Values of h greater than 0.5 are typically interpreted as relative to basin elongation with increasing size".

What about "c" ?

*With the aim of making the section 4.4 more clear, we have only provided natural range for the constant "c", focusing on the meaning of "h" in terms of drainage network geometry.*

17) L482-484 : "Both ksn_MOD and ._MOD (Fig. 12) are generally comparable with data coming from natural compilations (e.g. Kirby and Whipple, 2012). The matching of ksn and . between models and nature supports future development and application of the analog materials tested in this study for modeling landscape evolution."

What is written here is of course really important and should be strengthened to become more convincing for the audience. Presently, this comment is certainly "too short" to convince. Values of ksn or . could be quoted for instance. The geological contexts of selected natural examples could be also mentioned.

*We agree with the Reviewer. Now Figure 12 includes field data. Natural compilations are also proposed in the text (e.g. Kirby and Whipple, 2012).*

17) Figure 7: For CM3 and CM5, it is surprising to see the T2 profile above the T1 profile.

*We thank the Reviewer for pointing out this error in the figure. The stream extraction coming from Topotoolbox also extracted a numerical artifact at the bottom of the box. We fixed these streams in the revised version of the manuscript.*

18) Figure 10: Phase I and II could be merged on a single diagram to emphasize the decrease in

sediment discharge rate between the two phases.

*We agree with the Reviewer's comment and we have changed the revised figure accordingly.*

*Last but not least, in the revised version of the manuscript we have edited every grammatical error/mistyping the reviewer has highlighted. We have also added the references proposed by the Reviewer.*

**References**
- Graveleau, F., Hurtrez, J., Dominguez, S. and Malavieille, J.: A new experimental material for modeling relief dynamics and interactions between tectonics and surface processes, Tectonophysics, 513, 68–87, doi:10.1016/j.tecto.2011.09.029, 2011.
- Kirby, E. and Whipple, K. X.: Expression of active tectonics in erosional landscapes, Journal of Structural Geology, 44, 54–75, doi:10.1016/j.jsg.2012.07.009, 2012.
- Strak, V., Dominguez, S., Petit, C., Meyer, B. and Loget, N.: Interaction between normal fault slip and erosion on relief evolution: Insights from experimental modelling, Tectonophysics, 513(1–4), 1–19, doi:10.1016/j.tecto.2011.10.005, 2011.

---

## Author Comment (AC2) · 15 Sep 2020

We would like to thank Dr. Michele Cooke for the corrections and the useful comments. They have been very helpful in improving and making our work clearer and more readable. We include hereafter (as supplement file), point by point, the reply (in dark red italic text) to all the Referee comments. We will also upload the revised version of our manuscript. We hope the manuscript is now ready for the publication in Earth Surface Dynamics.

Please also note the supplement to this comment: https://esurf.copernicus.org/preprints/esurf-2020-35/esurf-2020-35-AC2-supplement.pdf

[Figure]

**ESurfD**

Interactive
comment

**Supplement:**

[Figure]

**DIPARTIMENTO di SCIENZE**
Viale G. Marconi, 446
00146 - ROMA

**Riccardo Reitano**
**Univ. Roma Tre**

To: Dr. Michele Cooke
Referee, *Earth Surface Dynamics*

Reference: Response to referee comments to the manuscript *" Erosional response of granular material in landscape models"* [Paper esurf-2020-35] by R. Reitano, C. Faccenna, F. Funiciello, F. Corbi, S. D. Willett.

Dear Dr. Michele Cooke,

we would like to thank you for the corrections and the useful comments. We include hereafter, point by point, the reply (in *dark red italic* text) to all the Referee comments. We also upload the revised version of our manuscript. We hope the manuscript is now ready for the publication in *Earth Surface Dynamics*.

With my kindest regards,

Riccardo Reitano, on behalf of all authors.

**Referee – Michele Cooke**
The paper presents a nice parametric study for a range of potential analog materials that could be used in coupled deformation and erosion models. The careful study very nicely contributes to our knowledge of these materials and will be a very valuable resource for future studies. Delineating a material recipe that can effectively simulate both deformational and erosional processes within the same experiment will enable many future investigations into the fascinating coupled feedbacks between these processes. I have a few suggestions that may strengthen the paper. These suggestions primarily relate to the framing and presentation of the findings rather than the results themselves, which are nicely gathered.

**Main Comments**

1) The need for analog models of erosion presented within the introduction can be strengthened. A reader unfamiliar with analog approaches might not be convinced that these approaches are needed from reading the somewhat vague statement in the manuscript that computation capacities are limited (line 40). Can you provide some examples of the limits of computational models? One approach may be to follow the reasoning presented in Reber et al (2020) for the benefit of analog models over numerical approaches. But the authors may have other even more compelling reasons to offer the reader.

*In agreement with the Reviewer's suggestion, we have modified the text adding a part in which we clarify the reason why we are using analogue models, and which are the advantages as respect to numerical modelling (lines 40-44). We welcomed the suggestion by the Reviewer and implemented the text also citing the work from Reber et al. (2020).*

2) The study can benefit from stronger support for the performance assessment. The manuscript frames its primary goal as "finding an analogue material that best mimics the erosional behavior of the natural prototype." The part that is missing is how specifically is good behavior assessed. How 'should' a slope of 15 degrees respond to the simulated precipitation rate? How do we know what experimental response is correct or wrong? The text states that a proper concave upward river profile is desirable (line 309) but are there some plausible conditions that would yield more or less concavity to the profile? My, albeit very

basic, understanding of fluvial mechanics is that the concavity of river profiles relates to the different strength of channel bed materials along the profile. If the material is uniform along the river profile, what degree of concavity is expected? This information is very important for helping us assess the performance of the material. Can you add to the figure 7 the range of expected profile shapes? Could you use numerical solutions for similar slope and precipitation but at crustal scale and with a range of soil/rock properties to validate the performance of the analog materials? The comparison of the laboratory observations to numerical models may be more straight-forward than comparison to natural systems because the numerical models can have uniform properties and starting slope.

*We revised the manuscript following the Reviewer's comment. What we are searching for in this study is a material that creates the same morphologies of those observed in natural orogens when is subjected to a) a tilt of 15° (the choice of the angle comes also from previous works as indicated in the text at lines 209, 384); b) a rainfall rate between 25 and 30 mm/h; c) by the only gravitational force without any other external forcing. In particular, the main argument when defining a "good" or "proper" analogue material is the channelization. A proper material should display channelization, with branching river network. As highlighted in section 2.2.1, we have channelization when the ratio between precipitation rate and infiltration capacity induces the surface runoff that, together with the mechanical properties of the material (cohesion, internal friction angle), allows for the formation of channels. Moreover, as explained in sections*

*2.2.1, 4.4 and 4.5, the model must be consistent with Hack's Law and slope-area scaling. We applied changes to the revised version of the manuscript trying to clarify this goal (lines 80-81, 185-187, 403-407). Following the Reviewer's suggestion, we also changed Figure 7, adding the shapes where concavity (θ) is 0, 0.5 and 1. Due to the very big range of natural values for θ (and also $k_{sn}$), we think it would*

*be clearer for the reader to see the shapes of the theoretical profiles. As far as the potential strength of the comparison between analogue and numerical simulations is concerned, we really appreciated the Reviewer's comment, as this topic will be studied in a future work not only because time consuming but also because of its high importance. Here we attach a picture of the coupled models we are currently running.*

3) Presumably, each of the tested combinations of analog materials may be suitable for different conditions of geologic substrate or precipitation rate. Perhaps one recipe is best suited for 'typical' settings but others could be used if you want to look at some atypical geomorphic settings. Providing some guidelines for the conditions under which each recipe could be suitable (or not) would be helpful.

*We agree with the Reviewer's comment. It is true that a combination of those materials may lead to some typical geomorphological settings (narrow or wide channels, more or less incise etc.). The difficulties lay in the link between the applied rainfall and the developed morphologies. The CM1 experiment is a good example of this concept. The rainfall imposed for the experiments (including CM1) remarks wet regions on Earth. But the structures that we observe in our model CM1, can be developed in nature in arid regions. Anyway, other materials (or combinations of materials) would not be suitable for landscape evolution models, because they do not satisfy the criteria for "good" or "proper" analogue material highlighted before, even in different climatic conditions or geological settings. Unfortunately, such study would be very time consuming and specific for a given process, while we are focusing on mountain building and erosion.*

4) The comparison in section 4.5 of geomorphic parameters to measurements from crustal geomorphic systems and the limits for various processes is very interesting. Can you please add the values of K and theta for detachment limited and transport limited systems to figure 12? This will aid in the comparison.

The last sentence of this paragraph asserts the match of the parameters with nature. Please add the ranges for geomorphic estimates of K and theta to figure 12.

*In agreement with the Reviewer's comment, figure 12 displays now also natural values for $k_{sn}$ and $\theta$. These values have been selected from field data coming from different areas and data compilations.*

5) Just a small note on porosity. If I understand correctly, the porosity is measured my comparing the volume prior to and after shaking. Doesn't this presume that they packing has no voids after shaking? If so, this doesn't seem realistic within granular materials which cannot pack with zero porosity. Maybe I'm missing something about this particular method of porosity measurement.

*The approach used to measure the material porosity has been implemented to reproduce measuring conditions close to the experimental ones. We are not assuming that the packing has no voids after shaking, but we are measuring the porosity before and after the shaking. The selected porosity value is the one recorded once the porosity values reach a plateau. We confirmed these values measuring the weight of material water-saturated and after the oven drying.*

*Last but not least, in the revised version of the manuscript we have edited every grammatical error/mistyping the reviewer has highlighted. We have also added the references proposed by the Reviewer.*

**References**

- Reber, J. E., Cooke, M. L. and Dooley, T. P.: What model material to use? A Review on rock analogs for structural geology and tectonics, Earth-Science Reviews, 103107, 2020.

---

## Author Comment (AC3) · 15 Sep 2020

Dear Dr. Braun

Following the ESurfD form, I uploaded the revised version of the manuscript. As requested, I uploaded also a marked-up version of the manuscript in the same file where the responses to referee comments are included.

With my kindest regards, Riccardo Reitano, on behalf of all authors.